

# Dynamic changes of optical and chemical properties of tar ball aerosols by atmospheric photochemical aging

Chunlin Li,[†] Quanfu He,[†] Julian Schade,[|] Johannes Passig,[|,|] Ralf Zimmermann,[|,|] Alexander Laskin,[§] and Yinon Rudich[†,*]

[†]Department of Earth and Planetary Sciences, Weizmann Institute of Science, Rehovot 76100, Israel

[|]Joint Mass Spectrometry Centre, University of Rostock, Dr.-Lorenz-Weg 2, 18059 Rostock, Germany

[|]Joint Mass Spectrometry Centre, Cooperation Group 'Comprehensive Molecular Analytics' (CMA), Helmholtz Zentrum München, Ingolstädter Landstrasse 1, 85764 Neuherberg, Germany

[§]Department of Chemistry, Purdue University, West Lafayette, Indiana 47907, United States

*Correspondence to*: Yinon Rudich (yinon.rudich@weizmann.ac.il)


**Abstract.** Following wood pyrolysis, tar ball aerosols were generated from wood tar separated into polar and
nonpolar phases. Chemical information of fresh tar balls was obtained from a high-resolution time-of-flight
aerosol mass spectrometer (HiRes-ToF-AMS) and single-particle laser desorption/resonance enhanced
multiphoton ionization mass spectrometry (SP-LD-REMPI-MS). Their refractive index between 365 and 425
nm was retrieved using a broadband cavity enhanced spectroscopy. Dynamic changes of the optical and
chemical properties for the nonpolar tar ball aerosols in NOx-dependent photochemical process were
investigated in an oxidation flow reactor (OFR). Distinct differences in the chemical composition of the polar
and nonpolar tar aerosols were identified. Nonpolar tar aerosols contain predominantly high-molecular weight
unsubstituted and alkyl-substituted polycylic aromatic hydrocarbons (PAHs), while polar tar aerosols consist of
a high number of oxidized aromatic substances (e.g., methoxy-phenols, benzenediol) with higher O:C ratio and
carbon oxidation state. Fresh tar aerosols have light absorption characteristics similar to atmospheric BrC with
higher absorption efficiency towards the UV wavelengths. The average retrieved refractive index (RI) are
1.661+0.020i and 1.635+0.003i for the nonpolar and polar tar aerosols, respectively, with absorption Ångström
exponent (AAE) between 5.7 and 7.8 in the wavelength region 365~425 nm. The RI fits a volume mixing rule
for internally mixed nonpolar/polar tar aerosols. The RI of the tar aerosols decreased with increasing wavelength
under photochemical oxidation. Photolysis by UV light (254 nm), without strong oxidants in the system, slightly
decreased the RI and increased the oxidation state of the tar balls. Oxidation under varying OH exposure levels
and in the absence of NOx diminished the absorption (bleaching), and increased the O:C ratio. The
photobleaching of tar ball aerosols via photochemically induced OH-oxidation is mainly attributed to
decomposition of chromophoric aromatics, nitrogen-containing organics, and high-molecular weight
components. Photolysis of nitrous oxide ($N_2O$) was used to simulate NOx-dependent photochemical aging of tar
balls in the OFR. Under high NOx conditions, photochemical aging lead to the formation of organic-nitrates,
increased oxidation degree and increased light absorption for the tar ball aerosols. These observations suggest
that secondary organic nitrate formation compensates the bleaching by photolysis and OH radical
photooxidation to eventually regain some absorption of aged tar balls aerosols. The atmospheric implication and
climate effects from tar balls upon various oxidation processes are briefly discussed.



## 1 Introduction

Organic aerosol (OA), which represent a ubiquitous and dominant burden of the tropospheric particulate pollutants, play important roles in atmospheric chemistry and balance of regional and global radiation (Jimenez et al., 2009; Kanakidou et al., 2005; Seinfeld and Pandis, 2016; Shrivastava et al., 2017). An indirect climate influence of OA relies on their interaction with water thus acting as cloud condensation nuclei (CCN) that may alter the hydrological cycle (cloud formation and perception) and modify Earth's albedo (Forster and Taylor, 2006; IPCC, 2013; Seinfeld and Pandis, 2016). The direct climate effect of OA is through extinction of incoming solar radiation and outgoing longwave radiation. Of particular importance is the warming effect due to light-absorbing carbonaceous aerosol commonly termed as brown carbon (BrC) (Andreae and Gelencsér, 2006). BrC is an important yet poorly understood OA component due to its complex physical properties, undefined chemical composition, and also its dynamic evolution under atmospheric processes (Adler et al., 2010; Moise et al., 2015; Laskin et al. 2015). It has been estimated that BrC accounts for 10-40% of the total light absorption in the atmosphere and when deposited on snow pack (Bahadur et al., 2012; Park et al., 2010), and contributes to global forcing of 0.10-0.25 W m$^{-2}$, with even higher values on regional scales (Feng et al., 2013).

The origin of BrC can be either primary (i.e., directly emitted) or secondary (i.e., generated by reactions of aromatic or carbonyl compounds in clouds or particles) (Laskin et al., 2015). On a global scale, biomass burning releases over two-thirds of primary BrC and also contributes substantially to overall secondary OA formation (Jacobson, 2014; Jo et al., 2016). Better understanding of the optical properties of biomass burning BrC aerosols is crucial for constraining its atmospheric and climatic implications and Earth's energy balance. Unlike black carbon that absorbs light strongly throughout the entire UV-visible range, different chromophores that may also be coupled via charge transfer complexes enable BrC absorption in a much more pronounced and complicated wavelength-dependence manner (Phillips and Smith, 2004; Reid et al., 2005; Lin et al., 2016, 2017).

Tar balls are a specific type of particles produced from wood combustion (especially from biomass smoldering burning) which are abundant in the troposphere (Pósfai et al., 2004; Hand et al., 2005; Chen et al., 2017). Tar ball particles have been collected and identified in many biomass burning plumes (Pósfai et al., 2004; Fu et al., 2012; Li et al., 2017). Microanalysis has found that tar balls are homogeneous spherical carbonaceous particles with sizes ranging from tens to hundreds of nanometers. These particles contribute a considerable fraction of the biomass burning BrC (Pósfai et al., 2004; Hand et al., 2005; Li et al., 2017). The estimated burden of tar balls on regional and global climatic forcing has been emphasized. Tar balls from different burning



conditions and bio-fuels coexist with many other types of particles (e.g., inorganic salts, soot, and other
carbonaceous aerosols in form of internal or external mixing), and these smoke particles undergo rapid
atmospheric processing once they are released from the fire (Pósfai et al., 2004; Hand et al., 2005; Li et al., 2015,
2017). However, *in situ* determination of the optical properties of these particles has not been reported due to
inherent difficulty in selective tar balls sampling out of complex particle ensembles typical of field burning
emissions.

The complex refractive index (*RI=n+ki, n* and *k* are real and imaginary parts, corresponding to scattering

and absorption, respectively) is an intrinsic optical property of aerosols. Quantifying the RI of OA is highly
needed for evaluating the related radiative forcing influence (Moise et al., 2015). Recently, several studies have
investigated the optical properties of tar ball particles (Chakrabarty et al., 2010; Hoffer et al., 2016;
Sedlacek et al., 2018). The optical measurements reported for tar balls or other biomass burning BrC, were
discrete over several wavelengths that were constrained by instruments measuring particle light coefficients, or
indirectly inferred from calculations based on their electron energy-loss spectra or from UV-Vis absorption of
solutions containing dissolved tar balls (Alexander et al., 2008). Hand et al. (2005) measured light scattering
coefficients of tar balls-dominated fire plumes using a nephelometer, and reconstructed the scattering
coefficients with simplified organic carbon (OC) and elemental carbon (EC) data to get an average RI of
1.56+0.02i for tar balls at λ = 632 nm. Chakrabarty et al. (2010) measured the RI of tar ball particles from
smoldering biomass combustion at 405, 532, and 780 nm, they observed a clear wavelength-, biofuel-, and even
burning condition-dependent RI. The light absorption by tar balls was similar to humic-like substance (HULIS)
with an imaginary part (0.002~0.015) that increased exponentially towards the near UV wavelengths. Recently,
Hoffer et al. (2016) generated tar ball particles from flameless wood pyrolysis in the laboratory. They reported a
higher RI value of 1.84+0.2i at 550 nm, which fell closer to RI of soot than to that of HULIS. Large
discrepancies reside in these results and discrete RI values make it difficult to decipher the complicated
wavelength-dependence character of tar balls optical properties, which finally constrains the assessment of its
radiative forcing effect.

Freshly emitted smoke BrC contain chromophores with diverse chemical structures, polarity, and

volatility (Lin et al., 2016, 2017). After emission into the atmosphere, smoke particles undergo dynamic changes
as a result of dilution, coagulation, and chemical processing on scales of seconds to days, which eventually
affect the physiochemical properties of BrC particles during its lifetime in the atmosphere (Reid et al., 2005; Li
et al., 2015; Laskin et al., 2015). Sumlin et al. (2017) simulated atmospheric photooxidation of biomass burning





BrC, and reported that photooxidation diminishes their light absorption. Zhong and Jang (2014) investigated the
influence of humidity and NOx presence in photochemical aging of biomass burning BrC. They found that
sunlight faded the color of BrC, and humidity facilitated the decay of light absorption by BrC, while presence of
NOx delays the fading. Overall, they concluded that light absorption by BrC is governed by chromophores
formation and bleaching by sunlight in the atmosphere. Therefore, evaluating the climatic impacts of tar ball
particles requires more extensive investigation of its optical properties, and understanding of the dynamic
transformations of the optical properties during atmospheric aging.
In this study, we generated proxies for tar ball particles by flameless wood pyrolysis (Tóth et al., 2014; Hoffer
et al., 2016). This method allows consistent and continuous generation of tar ball proxy aerosols for studying
their properties and processes. The RI of the tar aerosols as a function of wavelength in the ultraviolet-short
visible region (365~425nm, 0.5 nm resolution) was determined using a broadband cavity enhanced spectrometer
(BBCES). A high resolution time of flight aerosol mass spectrometer (HiRes-TOF-AMS) and a single-particle
mass spectrometer applying laser desorption/resonantly enhanced multiphoton ionization (SP-LD-REMPI-MS)
were used for probing the chemical profile of tar ball aerosols under NOx-dependent multiple-day
photochemical oxidation. Specifically, the dynamic changes of their optical properties in correlation with their
chemical composition were investigated. The atmospheric implications and climate forcing due to atmospheric
aging of tar aerosols and evolution of their optical properties were also explored.
**2 Experiment**
**2.1 Tar ball particle generation**
Following the formation mechanism in biomass burning process, polydisperse tar ball particles were generated
from droplets of wood tar in the laboratory (Tóth et al., 2014; Hoffer et al., 2016). In this study, a similar
procedure was applied for producing tar ball aerosols. In brief, commercial wood pellets (Hallingdal Trepellets,
water content of 6.55 wt.%, size of 2~3 cm in length, 0.2-0.3 cm in diameter) were smashed, heated and
dry-distilled in absence of air (25ºC min$^{-1}$ increase to 530 ºC from room temperature, and held for 20 min at
530 ºC) to produce liquid tar-water emulsions (~25 mL per hundred grams of used wood pellets). The emulsions
were filtrated using 0.45 μm pore size filter to remove particulate matter or solid precipitation. After overnight
static stabilization, the wood tar solution was phase-separated into water soluble and non-soluble oily phases at
an initial 3:1 volume ratio. Herein, we will term these two fractions as 'polar' and 'nonpolar' phases,
respectively. The phase-separated solution was further concentrated using a heating plate at 300 ºC with N$_2$



purge flow to prevent oxidation. A final 1:1 volume ratio of polar to nonpolar phase was obtained; then the
concentrated solutions were sealed and stored in the dark under 2 °C for following experiments. With respect to
their potential reactivity and instability, the distillation products were used within a few days.

For optical properties measurement experiments, tar aerosols were produced by atomizing solutions of

diluted tar in methanol (Gradient grade for HPLC, purity> 99.99 wt.%, Merck) with high-purity $N_2$ as a carrier
gas. As the chemical composition for ambient tar balls is still unidentified, tar ball aerosols in this study were
generated from polar, nonpolar, and mixtures of two phase tar solutions at volume mixing ratio of 2:1, 1:1, and
1:2, respectively. Charcoal and quartz heating tubes (150 °C, residence time ~0.7s for particles at a nitrogen flow
of 1.0 LPM) were used after the atomizer to outgas the methanol from the gaseous and particulate phases. Mesh
filters (TSI) downstream were used to filter out ultrafine (less than 100 nm) particles.
**2.2 NOx-dependent OH oxidation of tar ball aerosols**
Heterogeneous oxidation of tar aerosols was simulated using an oxidation flow Reactor (OFR), shown
schematically in Figure 1. The OFR has been characterized (Kang et al., 2007; Peng et al., 2015, 2016) and the
operational procedures have been described previously (He et al. 2018). Briefly, the OFR consists of a
horizontal 13.3 L aluminum cylindrical chamber (46 cm long × 22 cm ID) operating in continuous flow mode.
The chamber is equipped with two power controllable ozone-free mercury-lamps with peak emission at λ= 254
nm (82-934-08, BHK Inc., CA, USA). The two UV lamps are surrounded by Pyrex sheath tubes that are
continuously purged with $N_2$ to cool the lamps and remove outgassing compounds. OH radicals in the OFR are
produced through photolysis of externally introduced $O_3$ under 254 nm illumination and the further reaction of
singlet oxygen ($O^1D$) with water vapor:
$O_3 + h\nu(254\,nm) \rightarrow O^1D + O_2$                                                   R1
$O^1D + H_2O \rightarrow 2OH$                                                                          R2

External $O_3$ was produced by irradiation of 0.2 LPM high purity $O_2$ using a mercury lamp (λ=185nm,

78-2046-07, BHK Inc., CA). The $O_3$ concentration downstream of the OFR was measured with an $O_3$ monitor
(2B Technology). A Nafion membrane humidifier (Perma Pure LIC, NJ) was used to supply water vapor to the
OFR. Tar ball aerosols carried by 1.0 LPM $N_2$ flow from the atomizer were introduced into OFR. The initial
aerosol concentrations in the OFR were mediated by controlling the concentration of the tar solution in the
atomizer until the concentration of 350 nm particles was above 100 $cm^{-3}$, as shown in Figure S1 of tar ball
aerosols size distribution (SI, supporting information). Therefore, a total flow of 5.5 LPM with 36~38% RH,



initial 27~28 ppm $O_3$, and 200~250 µg m$^{-3}$ tar ball particles (assuming material density of 1.0 g cm$^{-3}$) was
maintained, with a corresponding plug flow residence time (RT) of 144s for aerosols in the OFR.
The extent of simulated daytime oxidation by OH exposure was varied by changing the UV light intensity.
Here, OH exposures in the OFR were inferred by measuring the decay of added $SO_2$ (using Thermo $SO_2$
analyzer, model 43i) due to reaction with OH radicals at specific UV lamp intensity. A low concentration
(~60ppb) of $SO_2$ was used to minimize its influence on the OH radical reactivity. Typical total OH exposures
ranged from $(8.7\pm2.3)\times10^{10}$ to $(8.6\pm1.7)\times10^{11}$ molec cm$^{-3}$ s or 0.5~7 equivalent daytime atmospheric oxidation
days (EAD) were maintained, taking typical ambient average OH concentration as $1.5\times10^6$ molec cm$^{-3}$ (Kang et
al., 2007; Peng et al., 2015, 2016).
In addition to reactions with oxidants, organic aerosols may change its chemical and physical properties by
photolysis (Epstein et al., 2014; Lee et al., 2014; Wong et al., 2014). Therefore, the influence of light irradiation
during tar ball photochemical aging was assessed at the short exposure time in the OFR. Here, tar balls aging
was repeated at the same conditions (e.g., RT, RH, $N_2/O_2$ flow, tar balls concentration, UV lamp power) without
$O_3$ supply in the OFR. The 254 nm photon flux at specific to maximal UV lamp power was calculated by fitting
the OH exposure estimated from $SO_2$ decay and by the Aerodyne OFR Exposure Estimator (v3.1,
https://sites.google.com/site/pamwiki/hardware/estimation-equations).
Under polluted conditions, nitrogen oxides (NOx) are often involved in the atmospheric transformations of
organic aerosol and alter their physiochemical properties (Rollins et al., 2012; Ng et al., 2007; Lin et al., 2015).
Therefore, $NO_x$ influence on tar ball aerosol aging was also investigated. Due to rapid conversion of NOx
(NO+$NO_2$) into nitric acid ($HNO_3$) under high $O_3$ and OH concentrations, simple addition of NOx into OFR
cannot sustain NOx levels that compete with $HO_2$ radicals in the reaction with organic proxy (ROO). NOx
generated via $N_2O$ reaction with $O^1D$ has been modeled and tested to suit the characterization of NOx-dependent
SOA formation pathways using OFR (Peng et al. 2017; Lambe et al., 2017). In this study, $N_2O$ (99.999%)
addition of 0.5 vol.% and 2.0 vol.% were used during tar ball aerosol photochemical oxidation in the OFR, and
equivalent OH exposure of about 4.0 EAD was maintained. NOx (NO and $NO_2$) concentrations downstream of
the OFR was measured using a NO/$NO_2$ analyzer (Ecotech, Serinus 40 NOx). Experimental parameters
including initial $O_3$ and $N_2O$ concentrations, NOx, moisture ratio, maintained OH exposures and the
corresponding photon flux at 254 nm are presented in Table 1.
**2.3 Online optical and chemical characterization**





184 Prior to the optical measurements, excess ozone and NOx were removed from the sample air stream following

185 the OFR using two diffusion denuders packed with Carulite (Carus Corporation, Peru, IL) and one activated

186 charcoal tube. The stream flow was further dehydrated with two silica gel diffusion dryers. Afterward, the tar

187 ball aerosols were characterized by a combination of on-line chemical and optical instruments.

188 Bulk chemical fragments and organic elemental ratios of tar ball aerosols were monitored in real time by the

189 HiRes-ToF-AMS (Aerodyne Research Inc., Billerica, MA, USA) in alternating high sensitivity V and

190 high-resolution W modes. The working principles of the AMS have been described in details elsewhere

191 (DeCarlo et al., 2006). In short, aerosol particles are separated from the gas phase through an aerodynamic lens

192 system and then transferred into the vacuum system, where they are impacted onto a vaporizer at about 600 °C,

193 thus vaporizing the particles. The analyte vapors are ionized with 70 eV electron impact ionization (EI). A

194 time-of-flight mass spectrometer is used for high-resolution analysis of the ions. SQUIRREL v1.16 and PIKA

195 v1.57 codes (http://cires.colorado.edu/jimenez-group/ToFAMSResources/ToFSoftware/) were used to process

196 the collected AMS data. Four ion groups were classified as $C_xH_y^+$, $C_xH_yO^+$, $C_xH_yO_z^+$ (z>1), and $C_xH_yO_iN_p^+$ (i$\geqq$0,

197 p$\geqq$1) based on fragment features. The ions $O^+$, $OH^+$, and $H_2O^+$ were included in the $C_xH_yO_z^+$ group, as

198 concentrations of these species were calculated from the organic $CO_2^+$ ion abundance using the method in Aiken

199 et al (2008). The ambient improved (AI) atomic ratios of oxygen to carbon (O:C), hydrogen to carbon (H:C),

200 nitrogen to carbon (N/C), and organic mass to organic carbon (OM/OC) were generated from the measured ion

201 fragments.

202 Particle-bound organic molecules were measured using a custom single-particle time-of-flight mass

203 spectrometer. This instrument features laser desorption and resonantly enhanced multiphoton ionization

204 (SP-LD-REMPI-ToF-MS), allowing for the detection of aromatic substances on individual particles. Detailed

205 description and application of the instrument in LD-REMPI ionization mode is given by Bente et al. (2008) and

206 Passig et al. (2017). Briefly, aerodynamically accelerated particles are individually sized using laser velocimetry,

207 and heated by a pulsed $CO_2$ infrared laser (10.6 μm) to desorb organic molecules. Aromatic substances in the

208 gas plume are selectively ionized via REMPI by a KrF-excimer laser pulse (248 nm) and detected in the positive

209 MS flight tube. The REMPI-MS technique is very sensitive and selective for aromatic substances (Boesl et al.,

210 1978; Grotemeyer et al., 1986; Rettner and Brophy, 1981) and suitable for studies on pyrolysis and (wood)

211 combustion processes (Heger et al., 1999; Czech et al., 2017). For the tar ball aerosols it provides

212 complementary information to the HiRes-TOF-AMS spectra. A custom software on LabView basis records and

213 calculates the aerodynamic size and individual mass spectra of the particles.



For optical measurements, tar ball aerosols were size-selected using an Aerosol Aerodynamic Classifier (AAC,
Cambustion, UK). AAC has significant advantages over the commonly used Differential Mobility Analyzer
(DMA) classifier. The AAC classifies particles based on the aerodynamic size without charging and hence it
avoids the contribution of multiply charged particles, thus generating real monodisperse size-selected particles
distribution, reducing the errors associated with multiply charged large particles. In addition, the AAC has
higher particle transmission efficiency at the relevant size range (Tavakoli and Olfert, 2013, 2014).
Aerodynamic size-classified particles after the AAC were further scanned by a scanning mobility particle sizer
(SMPS, classifier Model 3080, DMA Model 3081, CPC model 3775, TSI) to derive their mobility size
distribution. The effective density of the tar balls was estimated assuming a shape factor of particle to be 1.0
(sphericity was verified using a scanning electron microscopy in this study):
$$\rho_{eff} = \frac{D_{aero}}{D_m}\rho_0 \qquad\qquad [1]$$
Where $\rho_{eff}$ is an effective density, $D_{aero}$ and $D_m$ are aerodynamic and mobility diameters, respectively. $\rho_o$ is unit
density of 1.0 g cm$^{-3}$.
Based on the derived effective density, size-specific tar aerosols covering mobility diameters between 175 to
350 nm with an interval of 25 nm were size-selected via AAC and introduced into a dual-channel broad-band
cavity enhanced spectrometer (BBCES) for light extinction ($\alpha_{ext}$) measurements in the wavelength of 360~395
nm and 385~435 nm (at resolution 0.5 nm). A detailed description of the instrument can be found elsewhere
(Washenfelder et al., 2013; Flores et al., 2014a, b). With the combination of a condensation particle counter (TSI
3575) to measure particle concentration ($N$) in series, size-specific particle extinction cross section ($\sigma_{ext}$) can be
calculated by Equation (2):
$$\sigma_{ext}(\lambda, Dp, RI) = \frac{\alpha_{ext}(\lambda, D_p, RI)}{N(D_p)} \qquad\qquad [2]$$
Where $\lambda$ is the wavelength of incidence light, $D_p$ is the particle mobility diameter.
Using the Mie-Lorenz scattering theory, the wavelength-dependent complex refractive index of spherical
homogeneous particles was derived (Pettersson et al. 2004; Abo Riziq et al. 2007). The retrieval algorithm was
limited to search for n≧1 and k≧0 as their physical boundaries. Thereafter, spectral dependent extinction,
scattering, and absorption cross sections ($\sigma_{ext}$, $\sigma_{sca}$, and $\sigma_{abs}$) were calculated from the complex RI at specific
particle size. Using these parameters, the single scattering albedo, indicating the scattering fraction of light
extinction (SSA = $\sigma_{sca}/\sigma_{ext}$), was calculated.
The absorption and extinction Ångström exponents ($Å_{abs}$ and $Å_{ext}$) describe the spectral dependence of aerosol



light properties, and are widely used in climate modeling (Russell et al., 2010). It is customary to extrapolate the
optical spectral absorption and extinction fitting to the range of wavelengths using a power law $\propto \lambda^{-\text{Åabs}}$ and
$\propto \lambda^{-\text{Åext}}$, respectively. In this work, we determined $\mathring{A}_{ext}$ and $\mathring{A}_{abs}$ with a linear regression of $\ln(\sigma_{ext})$ and $\ln(\sigma_{abs})$
against $\ln(\lambda)$ over the range of 365 to 425 nm:
$$\mathring{A}_{ext} = -\frac{\ln(\sigma_{ext})}{\ln(\lambda)} \qquad \mathring{A}_{abs} = -\frac{\ln(\sigma_{abs})}{\ln(\lambda)}$$ [3]
Here $\mathring{A}_{ext}$, $\mathring{A}_{abs}$, and SSA were calculated for tar ball aerosols with a median diameter of 150 nm.
**2.4 Offline optical characterization**
In addition to the *in-situ* measurements, tar ball particles were also collected quantitatively onto Teflon filters
(47 mm diameter, 0.45 μm porosity, Pall Corp.) at sampling flow rate of 2 LPM and then extracted using
methanol (HPLC grade, Merck) for offline UV-Vis absorption measurement (Cary 60 UV-VIS spectroscopy,
Agilent). Methanol extraction of organic compounds has been commonly performed in various studies (Hoffer
et al., 2006; Laskin et al., 2009; Yee et al., 2013; Finewax et al., 2018; Xie et al., 2017). Here we verified the
completeness of the extraction by extracting each filter twice with methanol. Moreover, vortex shaking (Vortex
Genie-2, Scientific Industries) rather than sonication was applied to avoid chemical degradation of the extracts
upon ultrasonic irradiation (Miljevic et al., 2014; Mutzel et al., 2013). The mass absorption cross section (MAC,
$m^2 \, g^{-1}$) and refractive imaginary $k$ of the dissolved tar ball materials were estimated based on following relations
(Chen and Bond, 2010):
$$MAC_{(\lambda)} = \frac{Abs_{(\lambda)} \times \ln(10)}{C \times b}$$ [4]
$$k_{(\lambda)} = \frac{\lambda \times \rho \times MAC_{(\lambda)}}{4\pi}$$ [5]
$Abs_{(\lambda)}$ is the base-10 absorbance result from UV-VIS spectroscopy (unitless), $b$ is the optical length of the
solution, $C$ is the mass concentration of solution ($g \, m^{-3}$), $\lambda$ is the incident light wavelength, and $\rho$ is material
density ($g \, cm^{-3}$). Here, the derived effective density $\rho_{eff}$ was used. The absorption Ångström exponent based on
MAC was also derived as $\mathring{A}_{abs\text{-}UV\text{-}Vis}$ over the 365-425 nm spectral range.
In addition, particles were impacted at a flow of 2.5 L min$^{-1}$ onto cyclopore track-etched polycarbonate
membrane (47 mm, 0.1μm porosity, Whatman Inc.) to investigate the morphology of tar balls using Scanning
Electronic Microscopy (SEM, JEOL JSM-7000F).
**2.5 Radiative impacts of tar ball aerosols**




To assess the climatic influence of tar ball particles, a wavelength-dependent direct shortwave aerosol simple
radiative forcing efficiency (SRF, W g$^{-1}$) was estimated using the clear sky air mass global horizonal solar
spectrum (AM1GH), assuming that tar ball aerosols form a uniform, optically thin aerosol layer at the lower
troposphere or on ground (Bond and Bergstrom, 2006; Levinson et al., 2010):
$$\frac{dSRF}{d\lambda} = -\frac{1}{4}\frac{dS_{(\lambda)}}{d\lambda}\tau_{atm(\lambda)}^2(1-F_c)[2(1-R_{sfc})^2\beta_{(\lambda)}MSC_{(\lambda)} - 4R_{sfc}\cdot MAC_{(\lambda)}]$$ [6]
Where $dS(\lambda)/d\lambda$ is the solar irradiance (photons s$^{-1}$ cm$^{-2}$), $\tau_{atm}$ is the atmospheric transmission (taking 0.79 for
simple calculation), $F_c$ is the cloud fraction (approximately 0.6), $R_{sfc}$ is the surface albedo (approximate 0.19 for
urban area ground and 0.8 for snow) (Chen and Bond, 2010), $\beta$ is the average up-scatter fraction (the fraction of
scattered sunlight that is scattered into the upward hemisphere), and $MSC(\lambda)$ is wavelength-dependent mass
scattering cross section, respectively. We simply calculated radiative forcing of particles with atmospheric
relevant size of 50 to 500 nm, and SRF was estimated and integrated over the measured range of 365~425nm.
The actinic flux over 365~425 nm was obtained from the "Quick TUV Calculator", available at
http://cprm.acom.ucar.edu/Models/TUV/Interactive_TUV/ using the following parameters: SZA (solar zenith
angle) of 0 degree, noon time, June 30, 2000, 300 Dobson overhead ozone column, surface albedo of 0.19 for
urban area and 0.8 for snow, and 0 km altitude.
**3 Results and discussion**
**3.1 Chemical composition and optical properties of fresh tar ball aerosols**
The chemical composition and optical properties of fresh tar ball aerosols generated from polar, nonpolar, and
mixtures of the two-phase tar solutions were characterized. The bulk mass spectra for polar and nonpolar tar ball
particles are shown in Figure 2. The mass spectra features and particle effective densities are summarized and
compared in Table S1. Distinct differences in the chemical composition were observed between polar and
nonpolar tar ball aerosols. The alkyl fragments ($C_xH_y^+$, e.g., $C_nH_{2n-1}^+$, $C_nH_{2n+1}^+$) dominate the signals for
nonpolar particles (accounting for ~56% of total fragments), implying that the nonpolar tar aerosols have
compositional similarity with common hydrocarbon organic aerosol (HOA). The $C_xH_yO^+$ fragments are the
primary ions for the polar tar aerosols, contributing ~42% of their mass spectrum, suggesting that most of the
organic constituents in the polar tar aerosols are substantially oxygenated. Both spectra exhibit significant
intensity at *m/z* 28 ($CO^+$), *m/z* 29 ($CHO^+$), and *m/z* 43 ($C_2H_3O^+$), indicating the presence of carbonyl ions. The
strong signal at *m/z* 31 ($CH_3O^+$) results from methoxy species that preferably partition into the polar tar fraction.



In addition, the significant signals at *m/z* 50-52 ($C_4H_2^+$, $C_4H_3^+$, $C_4H_4^+$), *m/z* 65 ($C_5H_5^+$), *m/z* 77-78 ($C_6H_5^+$, $C_6H_6^+$),
*m/z* 81 ($C_6H_9^+$), and *m/z* 91 ($C_7H_7^+$), which are characteristic of aromatic compounds, indicate that tar aerosols,
especially from the nonpolar phase, contain a considerable amount of aromatic organics or present high
aromaticity. Ion peaks at *m/z* 77-78, 81, and 91 are typical for monocyclic aromatics such as alkyl-substituted
benzene (for *m/z* 77-78, 91) and heterocyclic aromatics (for *m/z* 81) (Li et al., 2012). While the relative higher
signal at *m/z* 128 ($C_{10}H_8^+$) in the nonpolar tar aerosols mass spectra can be assigned to molecular ion of
naphthalene (Herring et al., 2015). Moreover, signals at *m/z* 55 and 57 ($C_3H_3O^+$, $C_3H_5O^+$) are signature
fragments of aliphatic and non-acid oxygenated organics that are used to trace cooking emissions (He et al.,
2010), and these two fragments were also observed in the tar ball aerosols. Similar to ambient biomass burning
emissions, $C_2H_4O_2^+$ (*m/z* 60) and $C_3H_5O_2^+$ (*m/z* 73), two characteristic fragments from levoglucosan and similar
cellulose pyrolyzed species (mannosan, galactosan) were detected in all the tar aerosols, and these fragments
were more prominent in the polar aerosols due to the solubility of levoglucosan and analogs in water. Weimer et
al. (2008) reported the percentage of *m/z* 60 and 73 for the burning of various woods to be 0.6-4.1% and
0.1-2.0%, respectively. The percentage of these two fragments in the tar aerosols (0.7-1.6% for *m/z* 60 and
0.5-0.9% for *m/z* 73) are comparable to the literature data, although the fuel and the pyrolysis procedure are
different. The *m/z* 137 peak is dominated by fragments of $C_8H_9O_2^+$ and $C_7H_5O_3^+$, and these fragments have been
determined in biomass burning emissions and were assigned to lignin-related ions with methoxy-phenolic
structures (Li et al., 2012; Li et al., 2014). Phenols and methoxy phenols are prominent compounds, accounting
for 41% of the identified organic species, in primary BBOA (Schauer et al., 2001). The signal at *m/z* 137 is
much higher in the nonpolar phase aerosols (1.0% and 0.5% for nonpolar and polar tar balls, respectively), and
the fraction of fragment *m/z* 137 is consistent with reference values of 0.3-2.0% (Li et al., 2012). *m/z* 44 ($CO_2^+$),
a marker fragment of carboxylic acids has been parameterized as *f*44 (fraction of mass spectrum signal at *m/z* 44)
to present the oxidation degree of organic aerosols (Aiken et al., 2008; Ng et al., 2010). Higher *f*44 values
indicate more oxidized OA (OOA), while less oxidized OA is characterized by lower *f*44 (Schauer et al., 2001).
Peroxides can also produce $CO_2^+$ signal via extensive fragmentation in the AMS (Aiken et al., 2008). *f*44 has
also been shown to be linearly correlated with the elemental O:C ratio of OA (Aiken et al., 2008). In this study,
*f*44 for the nonpolar and polar tar aerosols are 1.9% and 2.4%, and the corresponding O:C ratios are 0.25 and
0.44. The higher O:C and H:C ratios explain the polarity of the polar tar ball aerosols. The simplified average
carbon oxidation state ($\overline{OSc} \approx 2O:C-H:C$) describe the oxidation level of particulate complex organic mixtures
(Kroll et al., 2011). The calculated $\overline{OSc}$ for the tar ball aerosols are -1.05~-0.76, which agrees well with reference



values of -1.0~-0.7 for primary BBOA (Kroll et al., 2011). These values are in the broad range of -1.7~-1.6 for
HOA and -0.5~0 for semivolatile OOA (Aiken et al., 2008). In addition, a small fraction of nitrogen-containing
organic compounds (NOC) was detected, with the $C_xH_yO_iN_p^+$ group contributing 1.6~3.6% of the tar balls mass
spectra with estimated N:C ratio below 0.01, which agrees with previously reported N:C values of 0.008~0.018
for biomass burning emissions (He et al., 2010). Biomass burning is an important source of NOC in the
atmosphere: alkaloid and nitro-aromatic constituents were detected to be abundant constituents of the NOC
(Laskin et al., 2009; Lin et al., 2017). Nitroaromatic compounds were also identified in urban fire emissions
(Bluvshtein et al., 2017; Lin et al., 2017). Although these compounds constitute a small fraction of the BBOA
mass, these chromophoric NOC species accounted for 50~80% of the total visible light absorption by the
extractable BrC (Lin et al., 2017).
Fragments larger than 100 amu ($f_{m/z>100}$) contribute a large fraction of the total organic signals for tar ball
aerosols, consistent with biomass burning emissions that contain a large fraction of high-molecular weight
compounds (Ge et al., 2012; Zhou et al., 2017). $f_{m/z>100}$ is 32% for the nonpolar tar ball aerosols, which is higher
than that of the polar particles (15%), demonstrating that the nonpolar tar balls consists of more high-molecular
weight organics. The measured effective densities for polar and nonpolar tar balls are 1.33 and 1.24 g cm$^{-3}$.
Chemical characteristics from AMS and densities for internal mixture tar balls follow the volume-linear mixing
of polar and nonpolar tar solutions.
A considerable aromatic fraction in the tar ball aerosols was confirmed by the LD-REMPI-MS measurement.
Figure 3 presents the mass spectra of aromatic substances obtained for each one exemplary polar and one
nonpolar particle, respectively. Aerodynamic size distributions for the detected tar ball aerosols are given in
Figure S2, and substances identified in the mass spectra are listed in Table S2. The features in the mass spectra
are consistent with the polarity of examined tar ball aerosols. The complex REMPI-spectrum shows rows of
intense unsubstituted and partially alkylated PAH peaks in the nonpolar tar balls, including naphthalene,
acenaphthylene, phenanthrene, pyrene, and in particular, the softwood combustion marker retene at $m/z$ 234 and
some possible derivatives (oxidized-retene at $m/z$ 250 with one oxygen addition, methyl-retene at $m/z$ 248 with
one methyl addition) (McDonald et al., 2000; Shen et al., 2012). Retene and some of the aforementioned PAHs
are also observed in the polar aerosols' mass spectra with lower intensities. In contrast, the polar tar ball REMPI
mass spectrum shows strong peaks from oxidized aromatics, more specifically, benzenediol and
methoxy-phenols (indicating e.g. catechol, guaiacol, acetovanillone, syringaldehyde, conifery aldehyde, etc.).
These results correspond to the large fractions of CHO$^+$ and CH$_3$O$^+$ fragments and high O:C ratios observed for



the polar tar aerosols via HiRes-TOF-AMS, and are consistent with the strong signals of typical aromatic
fragments observed in the nonpolar tar aerosols in Figure 2. The dominance of aromatic compounds in tar ball
chemical composition agrees well with previous work on BBOA (Schauer et al., 2001; Wei et al., 2015; Bente et
al. 2008, 2009; Czech et al. 2017). Biomass burning is a major source of environmental PAHs (alkylated PAHs,
oxygenated-PAHs, phenols, nitrogen-substituted PAHs, etc.) in both particulate and gaseous phases, and
extensive emissions of PAHs from incomplete combustion pose a great threat to ecosystem and human health
due to their carcinogenic toxicity (Li et al., 2017; Shen et al., 2013; Sigsgaard et al., 2015; Shrivastava et al.,
2017). Moreover, the primary PAHs can act as precursors that substantially contribute to ambient SOA or BrC
when involved in atmospheric photochemical aging, leading to profound climatic influence (Yee et al., 2013; Yu
et al., 2014; Lu et al., 2011; Zhang et al., 2012 ).

The complex refractive index (RI) of tar aerosols was retrieved under the assumption that the particles have

similar chemical composition and a spherical shape. The SEM images shown in Figure S3 confirm the spherical
morphology and homogeneous composition of the tar ball particles sampled in this study. Electron energy-loss
spectroscopy (EELS) spectra (not shown) indicate that the tar ball particles contain major C and minor O, which
is consistent with the AMS result and previous work (Pósfai et al., 2004; Chakrabarty et al., 2010). Continuous
spectral-dependent RI and SSA for tar ball were derived and are presented in Figure 4, RI results for tar ball
aerosol at mixing ratio of 2:1 and 1:2 are presented in Figure S4. Although scattering dominates the light
extinction, absorption in the UV and in the visible ranges was unambiguously identified for the tar ball aerosols,
with characteristic absorption similar to atmospheric BrC and HULIS (Hoffer et al., 2006; Bluvshtein et al.,
2017; Lin et al., 2017). The imaginary part, $k$, increases towards the UV range, presenting 0.02~0.03 difference
over the measured spectra range. $n$ for the nonpolar tar balls decreased from 1.673 at 365nm to 1.647 at 425nm,
which is almost parallel to the descending $n$ for the polar tar balls ranging from 1.651 at 365nm to 1.625 at
425nm. $k$ is 0.029~0.013 for the nonpolar tar ball for light wavelength of 365~425nm, while the imaginary part
for nonpolar aerosols is 0.007 at 365nm and zero at wavelength longer than 410 nm, indicating that there is no
detectable absorption or $k$ is lower than our detection limit. The nonpolar tar aerosols have stronger light
absorption and relative higher scattering abilities, resulting in a lower SSA compared with the polar tar aerosols.
The SSA increases towards the visible wavelength from 0.86 at 365nm to 0.90 at 425nm for nonpolar tar ball,
and the corresponding values are 0.95 to 1.0 for the polar tar balls.

The optical properties of aerosols relate to their chemical composition. Evidently, most of the PAHs identified

in the tar ball aerosols with high intensity have strong absorption between 350 and 450nm, as shown in Figure



S5 (Samburova et al., 2016), which coincide the range of tar ball absorption measured here, implying that PAHs
could be a dominate contributor to the absorption of fresh tar balls. Higher imaginary $k$ can be explained partly
by the larger proportion of PAHs as well as more high-molecular-weight organics present in the nonpolar tar ball
particles, as conjugated aromatic rings and phenols contribute to the major chromophores in the wood smoke
(Laskin et al., 2015). High-molecular weight organics may resemble HULIS that can form charge transfer
complexes (Phillips and Smith, 2004), that can absorb light at a longer wavelength range. The result is
consistent with the finding that higher molecular weight and aromaticity result in stronger absorption for
atmospheric BrC (Dinar et al., 2008). Moreover, the higher NOC content may also contribute to the
chromophores in the nonpolar tar aerosols.

The average RIs at 375 and 405 nm are $1.671+0.025i$ and $1.659+0.017i$ for nonpolar tar ball aerosols. The

corresponding RIs are $1.647+0.005i$ and $1.635+0.04i$ for the polar tar ball aerosols. The imaginary part $k$
retrieved from the BBCES data, though low, agrees well with $k$ values calculated from UV-Vis absorption of the
bulk solution. The MAC for the methanol extracts of the fresh tar ball particles is shown in Figure S6. The
absorption may be different for complex materials in the particulate and in the aqueous phases since parameters
such as shape factor, mixing state, together with artifacts from the optical instruments detection and data
retrieval methods can all affect the final optical results, while solvent-dependent extraction/dissolving efficiency
of chromophores or solvent effect (e.g., pH in water solution) may impact the solution absorption coefficient
(Huang et al., 2018; Lin et al., 2017). The light absorption coefficient of particulate BrC has been reported to be
0.7~2.0 times that of bulk BrC extracts by Liu et al. (2013).

The absorption Ångström exponent ($\mathring{A}_{abs}$) is often used to describe the wavelength-dependence of aerosol

light absorption with a value of nearly 1 for BC particles and values substantially larger than 1 indicating the
contribution from BrC (Reid et al., 2005; Chen and Bond, 2010). In this work, $\mathring{A}_{abs}$ for nonpolar and polar tar
ball particles ranges from 5.9 to 6.8 between 365 and 425 nm, which is consistent with values of 5.7~7.8
calculated from the bulk absorption measurement. The nonpolar tar balls have a lower $\mathring{A}_{abs}$. The difference in
$\mathring{A}_{abs}$ reflects the different chemical composition of chromophores in the particles, as inferred also from the AMS
data. Bluvshtein et al. (2017) reported relative low values of $\mathring{A}_{ext}$ (2~3) and $\mathring{A}_{abs}$ (4~6) over 300~650 nm for
ambient fire plume, which are likely affected by BC present in the smoke aerosol and also due to lower
wavelength dependence of aerosol absorption and scattering over the longwave visible range. Overall, the
broadband optical results for fresh tar ball aerosols are consistent with limited discrete measurements of
atmospheric BBOA as summarized in Table 2. It has also been found that the biomass fuel type, combustion



conditions, and atmospheric processing eventually affect the optical properties of BBOA. Lack et al. (2012)
modeled core-shell absorption for primary organic matter (POM) and BC from biomass burning. They found
that the imaginary part of the RI and MAC of POM at 404 nm were 0.007±0.005 and 0.82±0.43 $m^2$ $g^{-1}$,
respectively. Charkey et al. (2010) compared the optical properties of tar balls from smoldering combustion of
different biomass. Fuel-dependent imaginary RI for tar ball at 405 nm was 0.008~0.015 and $Å_{abs}$ over 405~532
nm was 4.2~6.4, which is in line with the $Å_{abs}$ value of 6~7 reported for BBOA derived HULIS (Hoffer et al.,
2006). Sedlacek et al. (2018) observed a weak absorption for wildfire produced tar balls with RI of 1.56 +0.02i.
Sumlin et al. (2018) simulated BrC formation under different pyrolysis temperatures. The BrC produced from
over 300 ℃ combustion has imaginary part of $k$ = 0.05~0.09 and real part $n$ of 1.59~1.68 at 375 nm, and RI at
405 nm is 1.57+0.03i, corresponded $Å_{abs}$ over 375~405 nm is 6.4~7.4.

Optical mixing rules can be used to estimate or explain the refractive indices of internally mixed substances,

and three mixing rules are commonly applied in climate models: molar refraction of absorption (Jacobson,
2002), volume-weighted linear average of the refractive indices (d'Almeida et al., 1991), and the
Maxwell-Garnett rule (Chýlek et al., 2000). The "linear mixing rule" and molar refraction mixing rules were
tested in this work for mixtures of tar ball particles against the retrieved optical data. Relevant data analysis
details are provided in the supporting materials (Table S3-S4, Figure S7-S11). It was found that both mixing
rules can predict the index of refraction for the polar/nonpolar tar balls, and values calculated based on "linear
mixing rule" fit better with the experimental data.

**3.2 Photooxidation of tar ball particles**

Aerosols have a wide range of atmospheric lifetimes from hours to days, during which they are involved in
various atmospheric processes, resulting in changes of properties (Reid et al., 2005; Rudich et al., 2007; Jimenez
et al., 2009). Therefore, we studied the effects of photochemical oxidation of the nonpolar wood-pyrolyzed tar
ball aerosols to investigate the physiochemical changes that can occur during tar balls' atmospheric lifetime.
Figure 5 presents the evolution of the wavelength-dependent RI and SSA as a function of the aerosols' O:C ratio
following $NO_x$-free photochemical aging in the OFR. The oxidation covers 0.7-6.7 EAD. Substantial decrease of
the RI and an increase of the SSA are correlated with an increase of the O:C ratio. Specific parameters are
summarized in Table S5. Light scattering as well as the absorption by the tar balls aerosol decrease with
increasing OH oxidation. The tar aerosols lose their scattering and absorption significantly up to 3.9 EAD aging.
The average RI decreased from initial 1.661+0.020i to 1.632+0.007i, and the corresponding average SSA



increased from 0.89±0.01 to 0.96±0.02. Then, the RI by tar balls persisted with enhanced oxidation, so that the
MAC values remained stable after 3.9 days oxidation (Figure S12) suggesting that all the photochemical-labile
chromophores were largely eliminated, while the remaining fraction still presented some light absorption.
Forrister et al. (2015) also observed a stable fraction of biomass burning BrC that had persistent absorption even
after long photochemical evolution time in the ambient environment. They suggested that the remaining
persistent fraction determines the background BrC levels. In our study, the O:C ratio for tar ball aerosols
increased continuously with photochemical oxidation, implying production of oxygenated constituents
(carboxylic, carbonyl compounds, etc.), and the interaction between these increasingly oxidized species coupled
with the relative stable intrinsic chromophoric structures (e.g., fused aromatic rings in Figure 3) in some
supermolecular structure that may explain the persistent absorption for aged tar ball aerosols (Dewar and Lepley,
1961; Desyaterik et al., 2013; Samburova et al., 2016). In addition, a balance between photobleaching of
intrinsic chromophores and photochemical formation of BrC via gas-particle transfer, as well as dynamic
gas-particle partitioning of chromophores and products of their photo-degradation should also be considered in
the overall absorption behavior for BBOA during photochemical processes.

The observed photooxidation bleaching is consistent with previous studies on atmospheric processes of BrC.

Sumlin et al. (2017) conducted multiple-day photochemical oxidation on primary biomass burning BrC aerosols
and observed that BrC losses its light absorption and scattering in the near-UV wavelengths by aging. Their
derived RI at 375 nm decreased from 1.59+0.03i for fresh emission to 1.50+0.02i after 4.5 EAD oxidation with
a corresponding O:C ratio increase from 0.34 to 0.40. Decrease in the overall BBOA absorption and scattering
was also detected *in-situ* following a one day evolution by Adler et al. (2011). They monitored an average RI of
1.53+0.07i and 1.54+0.04i for aerosols dominated by open fire and smoldering emissions, respectively, while
the RI decreased to 1.49+0.02i of the aged aerosols during the following day. Zhong and Jang (2014) reported
that light absorption of wood smoke BrC was modified by the photochemical process, owing to the production
of BrC from SOA formation and loss of BrC from photochemical bleaching of the chromophores. The total
MAC for the BrC eventually decreased by 19~68% within one day of aging. They proposed that bleaching
occurred by excitation of electrons through the absorption of sunlight via π-π* (UV and near UV illumination)
or n-π* (visible wavelengths irradiation) transitions. Then, the excited electrons disrupted the conjugated
structure of chromophores, leading to the fading of wood smoke color.

When tar ball aerosols were illuminated merely by 254 nm UV light at residence time of 144s, photolysis

occurred, weakly diminished their light absorption, in line with the extent of photon flux exposure. UV





irradiation similar to the O_3.9 condition slightly decreased the average RI to 1.649+0.018i, indicating that
photolysis played a minor role in tar aerosols bleaching and contributed to less than 15% of imaginary k changes
in NOx-free photochemical aging process. Even at full power of UV lamps in the OFR, the average RI
decreased by 0.012 and 0.005i for maximum photolyzed aerosols (SI, Table S6-S7, Figure S13-S16). As we also
did not observe detectable optical changes in blank tests upon exposure to $O_3$ under dark (SI, Figure S17-S18),
the bleaching of the tar balls in the OFR is mainly attributed to OH-initiated chromophores decomposition via
heterogeneous reactions, rather than to $O_3$ oxidation or photolysis.
These results indicate a fundamental relationship between photochemical processes and decreasing light
absorption and scattering in tar ball aerosol as a proxy system for biomass burning BrC aerosols. The changes
are a consequence of chemical composition changes in BrC. These changes occur on a time scale of one daytime
atmospheric aging. In Figure 6a, the H:C, OM/OC, $\overline{OSc}$, and particle effective density versus O:C ratio are
shown. Figure 6b presents the contributions of $C_xH_y^+$, $C_xH_yO^+$, $C_xH_yO_z^+$, and $C_xH_yO_iN_p^+$ groups to the tar balls
composition under a range of OH exposure conditions. Mass spectra features and densities of the tar ball
aerosols under various processes are summarized in Table S8. Increasing the OH exposure leads to continuous
increase of O:C and H:C ratios, and higher $\overline{OSc}$ for the tar balls aerosols. This result is consistent with Sumlin
et al. (2017), who reported that the O:C and H:C for BBOA increased by ~0.08 and ~0.03 after 4.5 EAD
photochemical oxidation, respectively. In this work, the measured O:C ratio increased from 0.25 to 0.38 after
maximum aging, while the H:C ratio increased by 0.07 from initial value of 1.55.
Other previous studies also depicted dynamic change of elemental ratios for SOA upon aging (Aiken et al.,
2008). The H:C ratio may either increase or decrease, depending on the precursor type and oxidation conditions.
Overall, O:C and H:C ratios changes relate to specific chemical processes or/and to gas-particle mass transfer
during aging of aerosols (Heald et al., 2010; Kim et al., 2014). The tar ball aerosols consist of mostly reduced
species ($\overline{OSc}$ <0), which oxidize primarily via oxidative formation of polar functional groups to the carbon
skeletons. In OH-initiated oxidation, functionalization includes OH/OOH function group addition and
COOH:Carbonyl groups formation that increase the net oxygen content in SOA (Kroll et al., 2011). Hydration
or polar functional groups addition to unsaturated C-C bonds may also increase the H:C ratio. Moreover,
fragmentation or evaporation also mediate the O:C and H:C ratios of SOA in further aging (Zhang and Seinfeld,
2013; Kim et al., 2014). We attribute the increase in H:C ratio to such oxidation mechanisms that involve bulk
species in the particles. As shown in Table S8, $f_{m/z>100}$ decreased continuously with aging. After 6.7 EAD
photooxidation, $f_{m/z>100}$ contributed only 21% of the total organic signals. The decrease of $f_{m/z>100}$ indicates that





fragmentation reactions are involved in the photochemical evolution and decomposition of high-molecular
weight compounds, thereby, reducing the size of the conjugated molecular system. The persistent high value of
$f_{m/z>100}$ after 6.7 EAD photooxidation imply that some high molecular weight compounds remained in the tar ball
aerosols, and continue to contribute to light absorption ether as individual chromophores or as charge transfer
complexes. Figure 6b shows that while $C_xH_y^+$ fragments deplete with OH exposure, $C_xH_yO^+$ and $C_xH_yO_z^+$
fragments increase, implying the formation of oxygenated moieties in the tar ball aerosols. In addition, a
decrease in the $C_xH_yO_iN_p^+$ fraction was measured from initial 3.6% to 1.9% after maximum oxidation. Ng et al.
(2010) suggested to use $f44(CO_2^+)$ $vs.$ $f43(C_2H_3O^+)$ triangle space as indication of OA sources and for estimation
of their degree of oxidation and volatility. The $C_2H_3O^+$ (less oxidized fragments) indicate aldehydes or ketones.
High $f44/f43$ ratio indicates low volatility and high oxidation level of SOA. Moreover, high $f44/f43$ and O:C
ratio are associated with increased hygroscopicity and possible CCN activity of OA (Hennigan et al., 2011;
Lambe et al., 2011). The $f44$ $vs.$ $f43$ in this study varied with photochemical aging and fell within the expected
range for ambient OOA, as shown in Figure 7. Increase of $f44/f43$ ratio with OH oxidation in Figure 6(b)
depicted the increase of carboxylic and/or peroxide compounds compared to carbonyl species in the tar balls,
which is consistent with the atmospheric evolution of ambient biomass burning plumes (Hennigan et al., 2011;
Canonaco et al., 2015).

To infer the possible chemical processes, detailed mass spectra of tar aerosols upon 6.7 EAD photochemical

oxidation were compared to a fresh ones (Figure S19). We found that decrease of alkyl/alkenyl chains (e.g.,
$C_nH_{2n-1}^+$, $C_nH_{2n+1}^+$) and aromatic ring structure fragments (e.g., $C_6H_5^+$, $C_6H_9^+$) contributed the prominent changes
in the $C_xH_y^+$ group, and relative higher $CO_2^+$ increment relative to $C_2H_3O^+$ explained the increase in the $f44/f43$
ratio. The decrease in the abundance of $C_2H_4O_2^+$ ($m/z$ 60) and $C_3H_5O_2^+$ ($m/z$ 73) is consistent with recent studies
that levoglucosan or similar species can decay in the atmosphere due to photochemical oxidation (Hennigan et
al., 2010). The pronounced decrease of intensity at $m/z$ 137 ($C_8H_9O_2^+$ and $C_7H_5O_3^+$) suggests that the
methoxy-phenol components in tar balls depleted substantially upon maximal aging.

In summary, photochemical oxidation by OH radicals destructed the aromatic rings and methoxy phenolic

structures, which are expected to be the primary chromophores in the tar balls. The NOC content and
high-molecular weight species also depleted via OH oxidation. These chemical changes upon OH oxidation may
explain the observed diminishing in light scattering and absorption upon photochemical aging.
**3.3 NO$_x$-dependent tar ball particles oxidation**



536 $N_2O$ was recently introduced for simulating $NO_x$-dependent SOA formation pathways in OFR under high $O_3$

537 concentration, as $O(^1D)+N_2O$ reactions can be applied to systematically vary the branching ratio of the $RO_2+NO$

538 reactions relative to the $RO_2+HO_2$ and/or $RO_2+RO_2$ reactions over a range of conditions relevant to atmospheric

539 SOA formation (Lambe et al., 2017). Here we introduced 0.5 and 2.0 vol.% $N_2O$ to investigate NOx-involved

540 daytime aging of tar ball aerosols in the OFR. The OH exposures were maintained for all these tests at about 4

541 EAD. The $NO_2$ concentrations downstream of the OFR was measured to be 96.1±1.3 and 528.3±6.2 ppbv. The

542 concentration of static NO can be neglected under these oxidation conditions. Figure 8 shows the mass spectra

543 features for fresh and aged tar balls reacted in the absence/presence of $NO_x$. Parameters including organic

544 elemental ratios and densities are summarized in Table S8. In general, tar aerosols oxidized under $N_2O$ addition

545 exhibit higher O:C and relative lower H:C ratios. $NO_y^+$ ($NO^+$ and $NO_2^+$) signals appear in the mass spectra and

546 the intensities of $NO_y^+$ display a positive trend with $N_2O$ concentration, together with an increase of oxygenated

547 fragments ($C_xH_yO^+$ and $C_xH_yO_z^+$). The normalized fractions of hydrocarbon fragments ($C_xH_y^+$) slightly

548 decreased. The signal ratio of $NO^+$ ($m/z$ 30) to $NO_2^+$ ($m/z$ 46) is used to distinguish organic nitrate from

549 inorganic nitrate. The signal from standard inorganic nitrate (e.g., $NH_4NO_3$) has a typical $NO_2^+/NO^+$ ratio of

550 0.485 using our AMS (detailed mass spectrum is shown in Figure S20), similar to previous studies (Zhou et al.,

551 2017). The fraction of $NO_y^+$ ($NO^+$ and $NO_2^+$) signals in the aged tar aerosols increased from 0% to 0.7% and

552 1.5% at 0.5 vol.% and 2.0 vol.% $N_2O$ additions, respectively. The corresponding values of $NO_2^+/NO^+$ ratio are

553 0.162 and 0.174, which are lower than that for inorganic nitrates. In addition, the contribution of $C_xH_yO_iN_p^+$

554 fragments increased from 1.9% to 4.4% and 4.5% over the course of aging. Therefore, we conclude that NOC

555 rather than inorganic nitrate formed in the $NO_x$-dependent photooxidation process, resulting in an overall

556 increase of N:C ratio from 0.010 to 0.012 and 0.015. Additionally, the density of tar balls weakly increased from

557 1.24 to 1.26 with 2.0% $N_2O$ addition.

558  Detailed changes in the mass spectra over the course of the experiment are shown in Figure S21. Ions

559 indicative of cyclolakyl fragments (e.g., $C_2H_3^+$, $C_3H_5^+$, $C_4H_7^+$) decreased, while $NO_x$ addition increased the $CO^+$

560 and $CO_2^+$ intensities relative to $C_2H_3O^+$, leading to higher $f44/f43$ ratio. Ng et al. (2007) observed a similar

561 change for photooxidation of terpenes in presence of $NO_x$. Changes of AMS spectra with $NO_x$ addition may

562 mark differences between the dominating reaction pathways in tar ball photooxidation as $RO_2+NO$ verses

563 $RO_2+HO_2$ and/or $RO_2+RO_2$ reactions.

564  Photochemical oxidation with $NO_x$ addition enhances the oxidation level and increases the absorption and

565 scattering of tar ball aerosols. Dynamic changes of the complex RI are shown in Figure 9 and summarized in



Table S5. The RI of tar ball aerosols increased from an average of 1.632+0.007i for pure OH-initiated
photooxidation to 1.635+0.015i with the addition of 0.5 vol.% $N_2O$, and a greater increase up to 1.648+0.019i
with 2.0 vol.% $N_2O$. The increase of RI is therefore primarily attributed to NOC formation. Zhong and Jang
(2014) observed that higher $NO_x$ level slowed photo-bleaching of wood smoke BrC, and they suggested that
NOx-modified reaction pathways produce secondary NOC chromophores (i.e., nitro-phenols). Liu et al. (2016)
simulated daytime chemistry of various VOCs in the presence of NOx and found that light absorption of
produced SOA, especially aromatic ones, increased with NOx concentration. These findings were also
corroborated by experimental study of Lin et al. (2015), where the chemical composition and the light
absorption properties of laboratory generated toluene SOA were reported to have strong positive dependence on
the presence of nitro-phenols formed at high NOx oxidation conditions. The color of the BrC diminished with
photolysis, correlated with a decline of the NOC fraction. Nitration of aromatic species via $NO_x$/$N_2O_5$/$NO_3$ has
been proposed as one of the main mechanisms to produce secondary BrC in the atmosphere (Lu et al., 2011; Lin
et al., 2015, 2017; Bluvshtein et al., 2017).
The imaginary part at 2.0 vol. % $N_2O$ addition was almost comparable with that of the fresh tar aerosols
(average value: RI=1.661+0.020$i$), although the real part was lower, suggesting that photooxidation in the
presence of $NO_x$ promote the formation of N-containing chromophores via secondary processes. In our
experiments, formation of the N-containing chromophores overweighed the bleaching from OH photooxidation
to eventually regain the absorption of aged tar balls. The SSA calculated for 150 nm particles decreased from
0.96 to 0.91 and 0.89 with $N_2O$ addition. Absorption enhancement with $N_2O$ addition for tar balls upon
photooxidation can also be seen in the MAC changes shown in Figure S22, where MAC at 375 nm for fresh tar
ball was 0.854 $m^2\,g^{-1}$, it decreased to 0.416 $m^2\,g^{-1}$ via OH photo-bleaching, then MAC increased to 0.459 $m^2\,g^{-1}$
at 0.5 vol.% $N_2O$ addition, and up to 0.598 $m^2\,g^{-1}$ at 2.0 vol.% $N_2O$ addition due to chromophores formation.
**3.4 Atmospheric and Climate implication**
Atmospheric aging alters the RI of SOA, and the dynamic changes of RI depend on complicated reaction
pathways (Liu et al., 2016). OH-initiated photochemical oxidation and photolysis decrease the RI of laboratory
proxies of tar balls under NOx-free condition, while photooxidation under high NOx has an opposite effect on
the RI of tar balls aerosol. We investigated the relationship between the dynamic RI values of tar ball particles
and their possible climatic implications, including the change of light extinction/absorption efficiency and the
clear-sky direct radiative forcing. For clarity, light extinction/absorption efficiencies were calculated and



compared at wavelength of 375 and 405 nm, while radiative forcing was estimated over all the measured
wavelengths from 365 to 425 nm. Atmospheric and climatic implications were assessed for fresh and oxidized
tar ball upon NOx-dependent ~3.9 EDA photooxidation (O_3.9, N_0.5, and N_2.0), in which fresh tar balls
were taken as reference.

As shown in Figure 10, photochemical oxidation under NOx-free condition (O_3.9) diminished light

extinction and absorption efficiency of tar balls aerosols in the atmospheric relevant size of 50-300 nm, causing
about 5~40% decrease in extinction at 375 and 405 nm wavelength. For aerosols larger than 400 nm, the
extinction efficiency of tar ball aerosols increased instead after photochemical aging. The light extinction
efficiency presented higher size-dependence than absorption, and extinction changes were more sensitive to
particle size, especially in the smaller sizes. The decreased absorption was more pronounced with ~60%
decrease at 375 nm and over 75% at 405 nm. Previous studies have confirmed the relationship between biomass
burning emissions and acute regional visibility deterioration (Huang et al., 2012; Chen et al., 2017). Our results
demonstrate that OH radical initiated daytime aging may play an important role in improving visibility
degradation caused by primary biomass BrC. However, photochemical evolution under high NOx conditions
may compensate effects of the photooxidation bleaching of tar ball aerosols via the formation of NOC
chromophores. At N_0.5 conditions, the light extinction decreased by 4 to 20% at 375 nm and 5 to 24% at 405
nm, respectively. The corresponding absorption decrease was 20~27% at both wavelengths. With more $N_2O$
addition, formation of secondary N-containing chromophores almost completely offsets light
extinction/absorption decrease caused by photooxidation. Under the N_2.0 conditions, enhancement of light
absorption efficiency for tar ball was about 0~9% at 405 nm in the entire size range of 50-500 nm.

Radiative forcing from aerosols over both ground and snow is vital to climate models (Barnett et al., 2005;

Kanakidou et al., 2005). Integrated radiative forcing for tar ball aerosols as a function of particle size under
various oxidation conditions is shown in Figure 11. Size-/wavelength-resolved SRF are also shown in Figure
S23 and S24. Integrated SRF over ground has negative values for tar balls over almost all the atmospheric
relevant sizes, indicating a radiative cooling effect by tar ball aerosols except at 195~210 nm, where fresh tar
ball particles present warming effect with SRF up to ~0.48 W $g^{-1}$. In practical fire emissions, the size of tar balls
depends on the burning and environment conditions and biomass fuel types with typical values between tens to
hundreds of nanometers (Reid et al., 2005; Pósfai et al., 2004). The complicated size-dependence character of
SRF makes it difficult to assess the real climatic effect of tar ball particles without extensive calculations. Figure
11(a) suggests fresh tar balls have SRF values of -7.46 W $g^{-1}$ at 150 nm and 0.45 W $g^{-1}$ at 200 nm, respectively.





The SRF decreased for all size ranges due to photochemical oxidation to -7.93 W g$^{-1}$ at 150 nm and -1.37 W g$^{-1}$
at 200 nm for tar ball aerosols under O_3.9 condition. At N_0.5 conditions, SRF was -7.37 W g$^{-1}$ at 150 nm and
0.16 W g$^{-1}$ at 200 nm, and the corresponding values at N_2.0 conditions increased to -7.20 W g$^{-1}$ at 150 nm and
0.31 W g$^{-1}$ at 200 nm.
In contrast, tar ball particles contributed to positive forcing (warming effect) over the bright terrain
throughout the atmospheric aging, as shown in Figure 11(b). Radiative forcing over the snow showed a simple
increasing trend with particle size, indicating that larger BrC aerosol with identical mass loading in the air have
a higher warming effect. The changes of snow-based radiative forcing upon photochemical aging followed the
same trends as in the ground-based cases. Fresh tar ball at size of 200 nm has SRF of 20.12 W g$^{-1}$ over the
incident solar wavelength of 365~425 nm on the snow terrain. With photochemical oxidation under NO$_x$-free
condition, radiative forcing decreased significantly. After 3.9 EAD atmospheric aging, snow-based radiative
forcing for tar ball decreased by 65~73% over the size range of 50~500 nm, the value of 200 nm tar ball became
6.99 W g$^{-1}$. When NOx was involved in the photochemical oxidation of tar balls, the decrement of radiative
forcing was weakened. At N_0.5, SRF for 200 nm tar ball was 14.01 W g$^{-1}$, while at N_2.0 condition,
size-dependent SRF from the aged tar ball was almost comparable with that from fresh tar ball, and SRF for 200
nm tar ball was 18.56 W g$^{-1}$.
Although less than 10% of the solar spectrum's energy is distributed between 365 and 425 nm, the radiative
forcing over this range represents a significant warming or cooling potential over the arctic terrain. In
conclusion, photochemical oxidation under NO$_x$-free conditions can decrease radiative forcing of tar ball
aerosols, resulting in enhancement in the cooling effect over ground and decreased in warming effect over the
snow. However, NOx involvement in photooxidation inhibits the decrease in radiative forcing of tar ball
aerosols. Overall, the complex changes in optical properties of tar balls at long aging times impose great
uncertainties in traditional model-based estimation of BBOA. Our study emphasizes the importance of taking
this atmospheric process into consideration to refine the understanding of the climatic and atmospheric
influences from these aerosols.
**4 Conclusions**
In this study, proxies for tar ball aerosols were generated in the laboratory following a flameless wood pyrolysis
process. The optical and chemical properties of the generated tar balls were constrained using BBCES and
HiRes-Tof-AMS/SP-LD-REMPI-MS and were shown to have many similarities to ambient biomass burning





aerosols. Laboratory generated fresh tar ball aerosols have light absorption characteristics similar to atmospheric
BrC with higher absorption efficiency towards the UV. The average complex refractive indices between 365 and
425 nm are 1.661+0.020i and 1.635+0.003i for nonpolar and polar tar ball aerosols, respectively.
Atmospheric evolution for tar ball aerosols was experimentally simulated using an oxidation flow reactor. The
study focused on dynamic changes in the optical and chemical properties due to NOx-dependent photochemical
oxidation. Furthermore, the relationship between oxidation level and the resulting RI of the tar ball aerosols was
explored. We found a substantial decrease in the scattering and absorption properties of tar balls, with a
corresponding increase in SSA with OH oxidation in the absence of NOx. A correlation between the RI decrease
and increase in the O:C and H:C ratios was observed. The decrease in light scattering and absorption is
attributed to the destruction of aromatic/phenolic/NOC and high-molecular weight species chromophores via
OH-initiated photooxidation of tar balls. Over longer aging times, the average RI of the tar ball aerosols
decreased from 1.661+0.020i to 1.632+0.007i upon atmospheric equivalent to 3.9 days aging, and the
corresponding O:C and H:C ratio increased from initial 0.25 and 1.55 to 0.35 and 1.59, respectively.
Our results suggest that OH oxidation rather than photolysis or ozone reactions plays the dominate role that
determine the optical and chemical properties in tar balls aging. The observed decrease in absorption results
from depletion of chromophores such as aromatic rings, phenolic compounds and high molecular weights
species.
Simulations under high $NO_x$ environment enhanced the aerosol oxidation state and increased the scattering
and absorption of tar ball aerosols relative to OH photooxidation in the absence of NOx. At ~3.9 EAD, addition
of 0.5 and 2.0 vol.% $N_2O$ increased the organic elemental ratios (O:C, H:C, and N:C ratios) and doubled the
organic nitrates fraction in the particles from 1.9 % to ~4.4 %. The formation of NOC chromophores overweigh
the intrinsic depletion of chromophores, leading to higher RI of 1.635+0.015i and 1.648+0.019i.
The atmospheric and climatic implications from tar ball aerosols under various oxidation conditions were
assessed using a simple radiative forcing model in terms of extinction/absorption efficiency changes and
ground-/snow-based radiative forcing. These results demonstrate that the optical and chemical properties of tar
ball particles are dynamically related to atmospheric aging, and optical changes are governed by both
photobleaching and secondary chromophores formation. Therefore, the atmospheric process should be
emphasized in model predictions for evaluating biomass burning BrC aerosol radiative forcing as well as
climate change.





**Acknowledgments**
This research was partially supported by research grants from the US-Israel Binational Science Foundation
(BSF) grant no. 2016093 and Israel Ministry of Science, Maimonide program. Dr. Li acknowledges support
from the Planning & Budgeting Committee, Israel (2018/19). J. Schade, J. Passig and R. Zimmermann
gratefully acknowledge financial support from the German Research Foundation, project number ZI 764/6-1,
and Photonion GmbH, Schwerin, Germany.





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





**Caption of Table and Figure**
**Table 1.** Summary of experimental conditions for tar balls photochemical oxidation
**Table 2.** Comparison of tar ball particle optical parameters with reference values of BBOA (mean ± standard
deviation)
**Figure 1.** Experimental setup for laboratory generation and aging of tar ball aerosol: including generation setup,
OFR photochemical aging, gaseous-particulate chemical monitoring, particle size distribution and optical
properties measurements.
**Figure 2.** High-resolution AMS mass spectra of fresh polar and nonpolar tar ball particles. Four ion groups are
grouped for clarity as: $C_xH_y^+$ (green), $C_xH_yO^+$ (purple), $C_xH_yO_z^+$ (z>1) (violet), $C_xH_yO_iN_p^+$(i≧0, p≧1) (light
blue). The mass fractions of the four fragment groups are presented by pie-charts.
**Figure 3**. LD-REMPI mass spectra of exemplary single tar ball particles, some feature peaks were identified
and labeled. a) Nonpolar tar ball spectra shows predominantly alkyl-substituted and unsubstituted PAHs. b)
Polar tar ball spectra reveals many oxidized aromatics, e.g., methoxy-phenol, benzenediol. Note the softwood
combustion marker retene at *m/z*=234, its characteristic fragments (*m/z*=203, 204, 205, 219) and possible retene
derivatives (*m/z*=248, 250).
**Figure 4.** Wavelength-dependent RI and SSA for tar ball particles generated from polar, nonpolar and mixture of
the two phases tarry solutions (only retrieval for mixture of 1:1 in vol. is shown for clarity, optical results for the
rest two mixtures can be found in supporting materials). The shaded areas indicate the upper and lower limits of
the imaginary part calculated from UV-Vis spectra of methanol extracts from the corresponding tar ball particles
samples: a) real part, b) imaginary part, and c) SSA calculated for 150 nm particles. Overplayed in green symbol
are previous measurements of biomass burning from the literature.
**Figure 5.** Evolution of the retrieved wavelength-dependent complex RI and SSA as a function of O:C ratio for
tar ball particles upon OH photochemical oxidation: a) real part, b) imaginary part, and c) SSA calculated for
150 nm particles. The color scale shows the span in the RI for the wavelengths measured from 365 to 425 nm.
For clarity, error bars for O:C ratio (±0.01), RI (±0.007 for real part, and ±0.003 for imaginary part on average),
and SSA (±0.006) are not shown. Two dashed lines trace the RI and SSA at 375 nm (purple) and 405 nm (green).
O_0.7~O_6.7 represent equivalent atmospheric photochemical oxidation for 0.7 and up to 6.7 days.
**Figure 6.** Dynamic changes for the chemical characteristics of tar ball particle under NOx-free OH
photochemical oxidation: a) OM/OC. H:C ratio, particle density, and average carbon oxidation state ( $\overline{OSc}$ )



changes as a function of O:C ratio; b) mass spectra evolution with oxidation times in term of $C_xH_y^+$, $C_xH_yO^+$,
$C_xH_yO_z^+$, and $C_xH_yO_iN_p^+$ fragment groups.
**Figure 7.** Comparison of $f44$ and $f43$ values from ambient data sets (Ng. et al., 2010) and values from ambient
biomass burning organic aerosol.
**Figure 8.** Dynamic changes for chemical characteristics of tar ball aerosols under $NO_x$-dependent OH
photochemical oxidation: a) OM/OC, O:C, H:C, and particle density changes; b) mass spectra changes with
different oxidation conditions in term of $C_xH_y^+$, $C_xH_yO^+$, $C_xH_yO_z^+$, and $C_xH_yO_zN_p^+$ fragment groups. $C_xH_yO_zN_p^+$
include all nitrogen-containing fragments, (e.g., $C_xH_yON^+$, $C_xH_yO_zN_i^+$, $C_xH_yN^+$, etc.), $NO_y^+$ include $NO^+$ and
$NO_2^+$. O_3.9 represents 3.9 days equivalent atmospheric photochemical aging in absence of NOx, N_0.5 and
N_2.0 indicate photochemical oxidation with 0.5 and 2.0 vol.% $N_2O$ addition at ~4.0 days atmospheric
oxidation.
**Figure 9.** Changes of retrieved spectra-dependent RI as a function of O:C ratio for tar ball particles upon
NOx-dependent photochemical oxidation: a) real part, b) imaginary part, and c) SSA calculated from 150 nm
particles. For clarity, error bars for O:C ratio (±0.01), RI (±0.006 for real part, and ±0.003 for imaginary part on
average), and SSA (±0.007) are not shown.O_3.9 represents 3.9 days equivalent atmospheric photochemical
aging in absence of NOx, N_0.5 and N_2.0 indicate photochemical oxidation with 0.5 and 2.0 vol.% $N_2O$
addition at ~4.0 days atmospheric oxidation.
**Figure 10.** Size-resolved light extinction and absorption efficiency ratio of $NO_x$-dependent photooxidized tar
balls compared to the fresh tar ball particles: a) and c) extinction ratios at 375 and 405 nm, b) and d) absorption
ratios at 375 and 405 nm. O_3.9 represents 3.9 days equivalent atmospheric photochemical aging in absence of
NOx, N_0.5 and N_2.0 indicate photochemical oxidation with 0.5 and 2.0 vol.% $N_2O$ addition at ~4.0 days
atmospheric oxidation.
**Figure 11.** Calculated size-resolved simple radiative forcing (SRF, W $g^{-1}$) by tar ball aerosols, integrated over
365~425 nm incident solar irradiation for fresh and NOx-dependent photooxidized tar balls: a) ground-based
radiative forcing, b) snow-based radiative forcing.





**Table 1**. Summary of experimental conditions for tar ball particles photochemical oxidation

| Experiment | $O_3$ (ppm) | $N_2O$ mixing ratio | Endpoint $NO_x$ (ppb) | RH (%) | water mixing ratio | Exposure | |
|---|---|---|---|---|---|---|---|
| | | | | | | OH radical (molecules cm$^{-3}$ s) | photon flux (photons cm$^{-2}$) |
| P1 | —— | —— | —— | 38.90 | 0.0126 | —— | 7.47E+15 |
| P2 | —— | —— | —— | 39.70 | 0.0128 | —— | 4.83E+16 |
| P3 | —— | —— | —— | 40.50 | 0.0130 | —— | 1.00E+17 |
| O_0.7 | 24.46 | —— | —— | 37.29 | 0.0120 | 8.68E+10 | 1.56E+15 |
| O_1.7 | 24.76 | —— | —— | 37.66 | 0.0122 | 2.23E+11 | 7.47E+15 |
| O_3.9 | 24.63 | —— | —— | 35.58 | 0.0115 | 5.11E+11 | 4.83E+16 |
| O_6.7 | 25.31 | —— | —— | 35.67 | 0.0116 | 8.65E+11 | 5.17E+16 |
| N_0.5 | 24.18 | 0.005 | 96.1 | 36.60 | 0.0118 | 5.37E+11 | 5.92E+16 |
| N_2.0 | 28.21 | 0.020 | 528.3 | 35.90 | 0.0116 | 4.85E+11 | 1.00E+17 |

Note: P1~P3 mean photolysis test, O_0.7~O_6.7 correspond to photochemical oxidation experiment from equivalent 0.7 day to 6.7 days
ageing, and N_0.5 and N_2.0 indicate photochemical oxidation with $N_2O$ addition at 0.5 vol.% and 2 vol.% mixing ratios (standard
deviation for the parameters were not given in above table)





**Table 2.** Compare of tar ball particle optical properties with reference values of BBOA (mean ± standard deviation)

| BrC | Complex Refractive index | | | $\text{Å}_{abs}$ | $\text{Å}_{abs\_UVVIS}$ | $\text{Å}_{ext}$ | Reference |
|---|---|---|---|---|---|---|---|
| | Average | 375nm | 405nm | | | | |
| Nonpolar | $(1.661\pm0.008)+(0.020\pm0.004)i$ | $(1.671\pm0.003)+(0.025\pm0.003)i$ | $(1.659\pm0.011)+(0.017\pm0.002)i$ | $5.87\pm0.37$ | 5.74 | $3.81\pm0.18$ | This work |
| Mixture (2:1 in vol.) | $(1.670\pm0.010)+(0.017\pm0.004)i$ | $(1.682\pm0.008)+(0.021\pm0.002)i$ | $(1.668\pm0.007)+(0.013\pm0.001)i$ | $6.79\pm0.91$ | 7.08 | $4.01\pm0.09$ | |
| Mixture (1:1 in vol.) | $(1.694\pm0.011)+(0.013\pm0.003)i$ | $(1.703\pm0.015)+(0.017\pm0.001)i$ | $(1.689\pm0.011)+(0.009\pm0.002)i$ | $6.16\pm0.54$ | 7.38 | $3.73\pm0.23$ | |
| Mixture (1:2 in vol.) | $(1.672\pm0.010)+(0.011\pm0.004)i$ | $(1.683\pm0.005)+(0.018\pm0.002)i$ | $(1.667\pm0.003)+(0.006\pm0.003)i$ | $6.66\pm0.63$ | 7.24 | $4.06\pm0.11$ | |
| Polar | $(1.635\pm0.009)+(0.003\pm0.003)i$ | $(1.647\pm0.003)+(0.005\pm0.001)i$ | $(1.635\pm0.004)+(0.004\pm0.003)i$ | $6.72\pm2.28$ [a] | 7.83 | $3.93\pm0.06$ | |
| BBOA | 1.590+0.029i@375nm, 1.570+0.010i@405nm (IPN) | | | 6.4~7.4 | | | Sumlin et al., 2017; 2018 |
| BBOA | 1.590+0.017i@405nm (IPN) | | | | | | Flowers et al., 2010 |
| BBOA | $k$: 0.009@404nm (CRDS-PAS) | | | | | | Lack et al., 2012 |
| Tar ball | 1.78+0.015i, 1.83+0.0086i@405nm (IPN) | | | 4.2~6.4 | | | Chakrabarty et al., 2010 |
| Tar ball | 1.56+0.02i @405nm (CRDS-UVVIS) | | | | | | Hand et al., 2005 |
| BBOA | 1.53+0.071i (WELAS,open fire), 1.54+0.04i (WELAS, smoldering) | | | | | | Adler et al., 2011 |
| BBOA | 1.64+0.03i@405nm (BBCES-Neph) | | | 4~6 [b] | | 2~3 [b] | Bluvshtein et al., 2017 |
| BBOA_HULIS | 1.653+0.002i, 1.685+0.002i@532nm(Nep-PAS) | | | | 6~7 | | Hoffer et al., 2006 |
| BBOA_HULIS | 1.616+0.023i@390nm(CRDS) | | | | | | Dinar et al., 2008 |
| BBOA | 1.550+0.033i@365nm (BBCES) | | | | | | Washenfelder et al., 2015 |
| BBOA | | | | | 6.9~11.4 [c] | | Chen and Bond, 2010 |
| BBOA | | | | | 5.3~8.1 [c] | | Xie et al., 2017 |
| Ambient SOA | | | | | 6.0~6.3 [c] | | Huang et al., 2018 |
| Ambient SOA | $k$: 0.046@365nm, 0.039@405nm, 0.036@420nm (LWCC) | | | | | | Shamjad et al., 2018 |

Note: $\text{Å}_{abs}$ and $\text{Å}_{ext}$ were calculated form tar ball particle with median diameter of 150 nm in this study
[a] regressed over wavelength range of 365~400 nm, no absorption detected over 410 nm using BBCES system
[b] regressed over wavelength range of 300~650 nm for bulk fire plume emissions
[c] $\text{Å}_{abs\_UVVIS}$ of methanol extracts over whole range from 300/360~600 nm
Instrument: IPN(integrated photoacoustic nephelometer), CRDS (cavity ring-down spectrometer), PAS (photoacoustic absorption spectrometer), WELAS (white light optical particle counter), LWCC (a liquid
waveguide capillary cell)





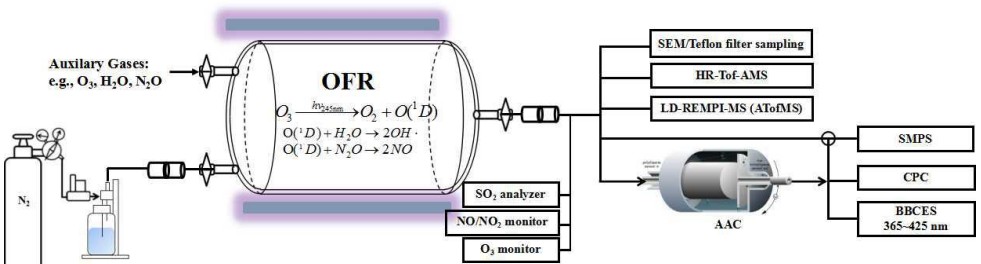


**Figure 1.** Experimental setup for laboratory generation and aging of tar ball aerosol: including generation setup, OFR
photochemical aging, gaseous-particulate chemical monitoring, particle size distribution and optical properties
measurements.





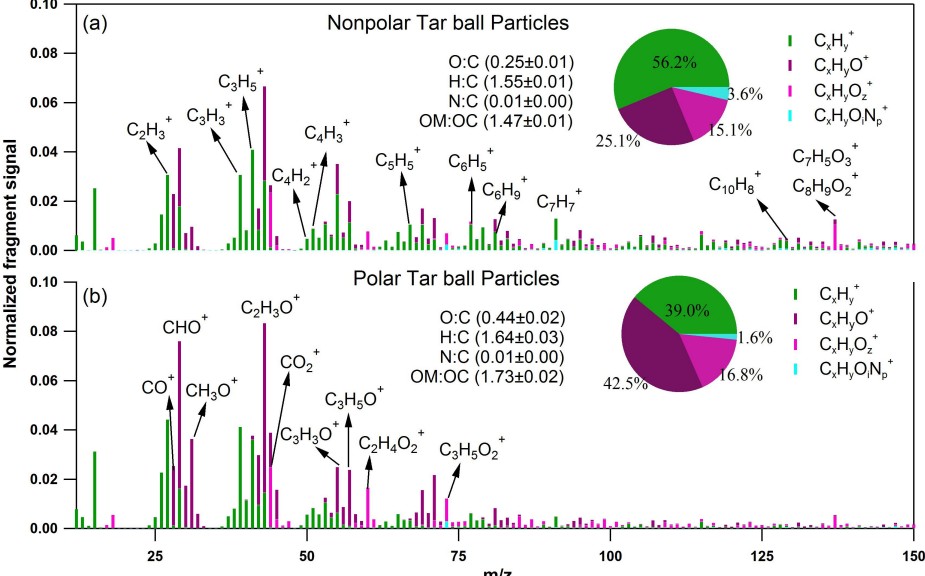


**Figure 2.** High-resolution AMS mass spectra of fresh polar and nonpolar tar ball particles. Four ion groups are grouped for

clarity as: $C_xH_y^+$ (green), $C_xH_yO^+$ (purple), $C_xH_yO_z^+$ (z>1) (violet), $C_xH_yO_iN_p^+$(i≧0, p≧1) (light blue). The mass fractions of

the four fragment groups are presented by pie-charts.





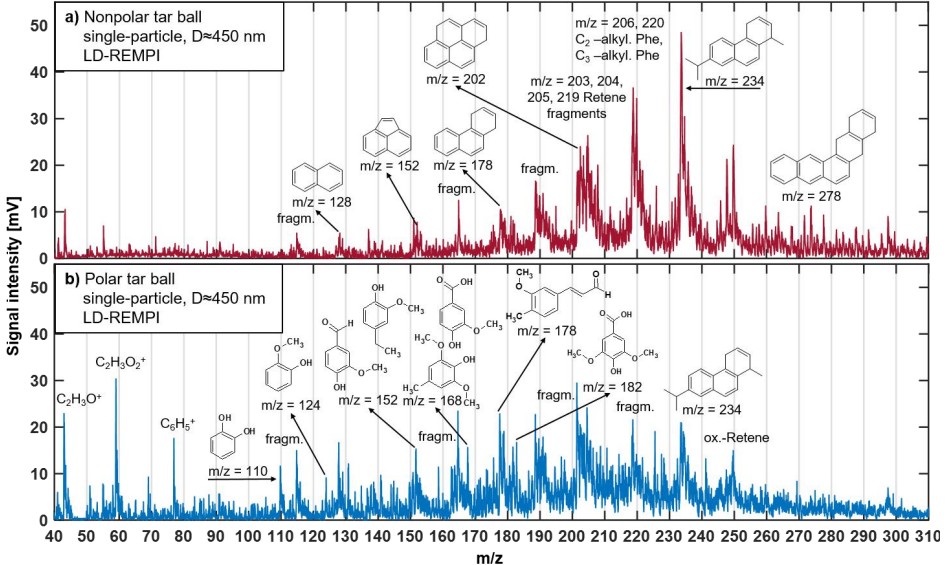


**Figure 3.** LD-REMPI mass spectra of exemplary single tar ball particles, some feature peaks were identified and labeled. a)

Nonpolar tar ball spectra shows predominantly alkyl-substituted and unsubstituted PAHs. b) Polar tar ball spectra reveals

many oxidized aromatics, e.g., methoxy-phenol, benzenediol. Note the softwood combustion marker retene at $m/z$=234, its

characteristic fragments ($m/z$=203, 204, 205, 219) and possible retene derivatives ($m/z$=248, 250).



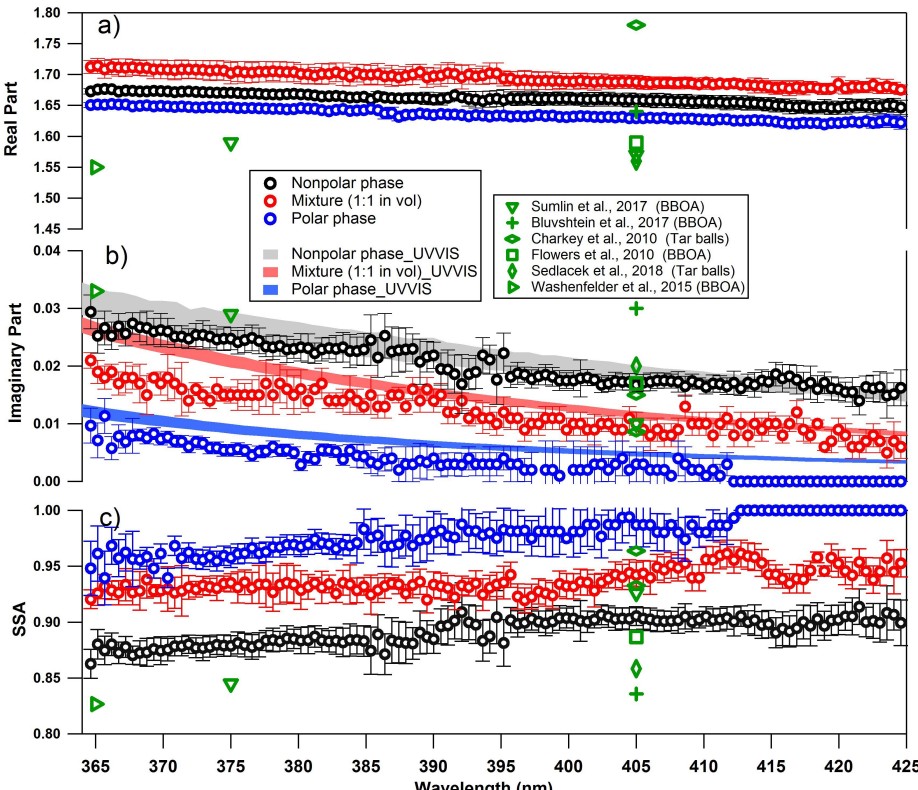

**Figure 4.** Wavelength-dependent RI and SSA for tar ball particles generated from polar, nonpolar and mixture of the two

phases tarry solutions (only retrieval for mixture of 1:1 in vol. is shown for clarity, optical results for the rest two mixtures

can be found in supporting materials). The shaded areas indicate the upper and lower limits of the imaginary part calculated

from UV-VIS spectra of methanol extracts from the corresponding tar ball particles samples: a) real part, b) imaginary part,

and c) SSA calculated for 150 nm particles. Overplayed in green symbol are previous measurements of biomass burning

from the literature.





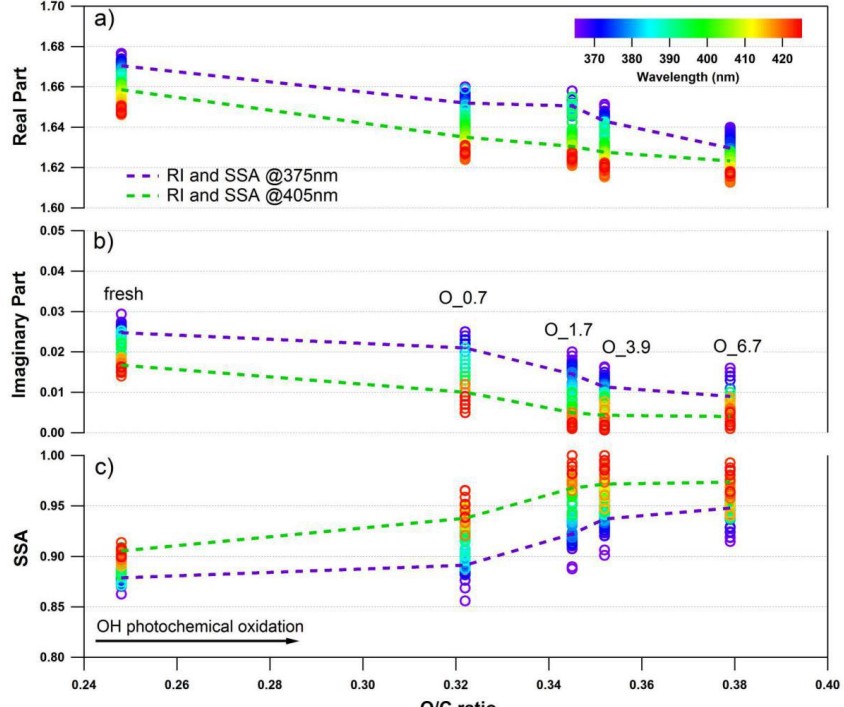

**Figure 5.** Evolution of the retrieved wavelength-dependent complex RI and SSA as a function of O:C ratio for tar ball

particles upon OH photochemical oxidation: a) real part, b) imaginary part, and c) SSA calculated for 150 nm particles. The

color scale shows the span in the RI for the wavelengths measured from 365 to 425 nm. For clarity, error bars for O:C ratio

(±0.01), RI (±0.007 for real part, and ±0.003 for imaginary part on average), and SSA (±0.006) are not shown. Two dashed

lines trace the RI and SSA at 375 nm (purple) and 405 nm (green). O_0.7~O_6.7 represent equivalent atmospheric

photochemical oxidation for 0.7 and up to 6.7 days.



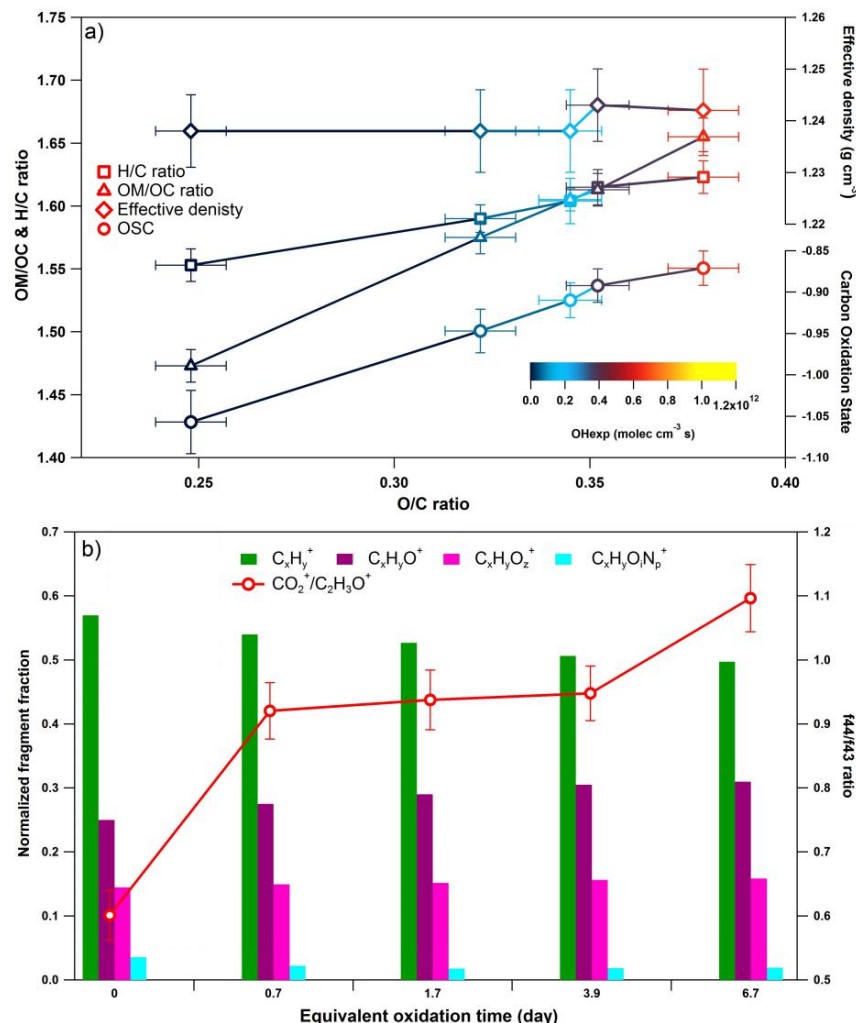

**Figure 6.** Dynamic changes for the chemical characteristics of tar ball particle under NOx-free OH photochemical oxidation:

a) OM/OC. H:C ratio, particle density, and average carbon oxidation state ($\overline{OSc}$) changes as a function of O:C ratio; b) mass

spectra evolution with oxidation times in term of $C_xH_y^+$, $C_xH_yO^+$, $C_xH_yO_z^+$, and $C_xH_yO_iN_p^+$ fragment groups.





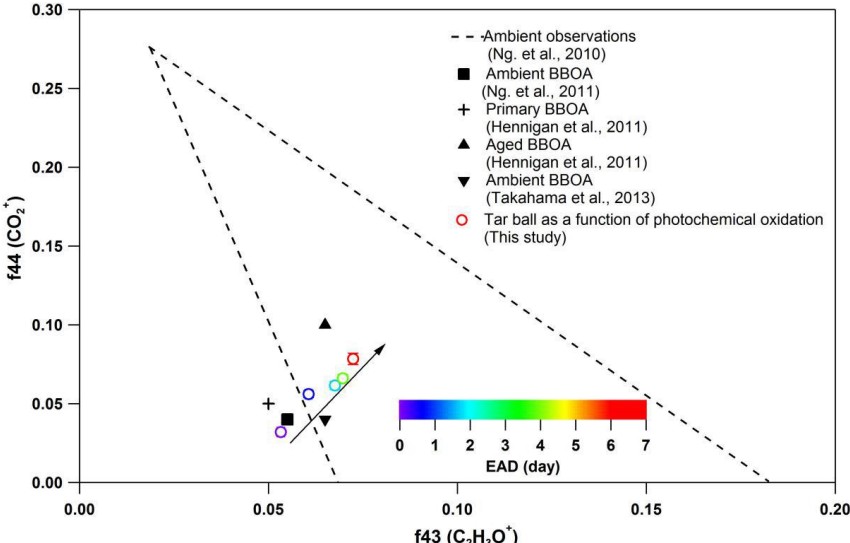

**Figure 7.** Comparison of f44 and f43 values from ambient data sets (Ng. et al., 2010) and values from ambient biomass

burning organic aerosol. The hollow circles present tar ball result in this work, and color legend indicate equivalent

atmospheric oxidation days, black arrow more clearly shows the extent of NOx-free photochemical oxidation in this study.





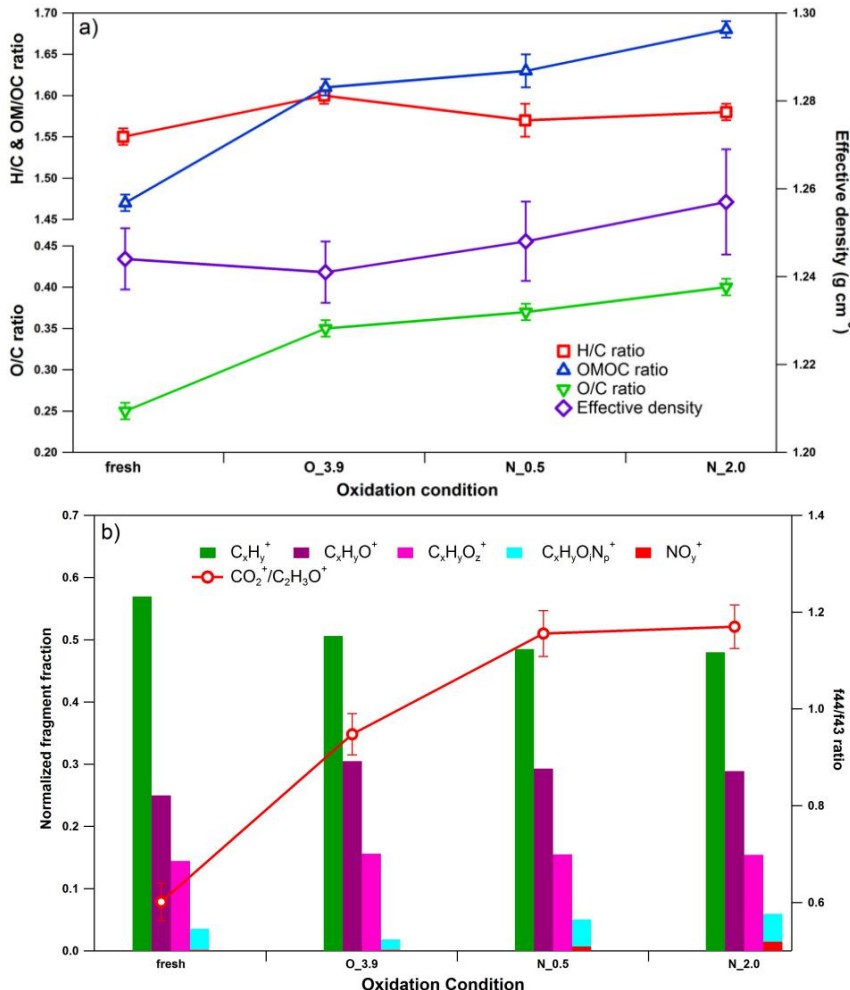

**Figure 8.** Dynamic changes for chemical characteristics of tar ball aerosols under $NO_x$-dependent OH photochemical

oxidation: a) OM/OC, O:C, H:C, and particle density changes; b) mass spectra changes with different oxidation conditions in

term of $C_xH_y^+$, $C_xH_yO^+$, $C_xH_yO_z^+$, and $C_xH_yO_zN_p^+$ fragment groups. $C_xH_yO_zN_p^+$ include all nitrogen-containing fragments,

(e.g., $C_xH_yON^+$, $C_xH_yO_zN_i^+$, $C_xH_yN^+$, etc.), $NO_y^+$ include $NO^+$ and $NO_2^+$. O_3.9 represents 3.9 days equivalent atmospheric

photochemical aging in absence of NOx, N_0.5 and N_2.0 indicate photochemical oxidation with 0.5 and 2.0 vol.% $N_2O$

addition at ~4.0 days atmospheric oxidation.



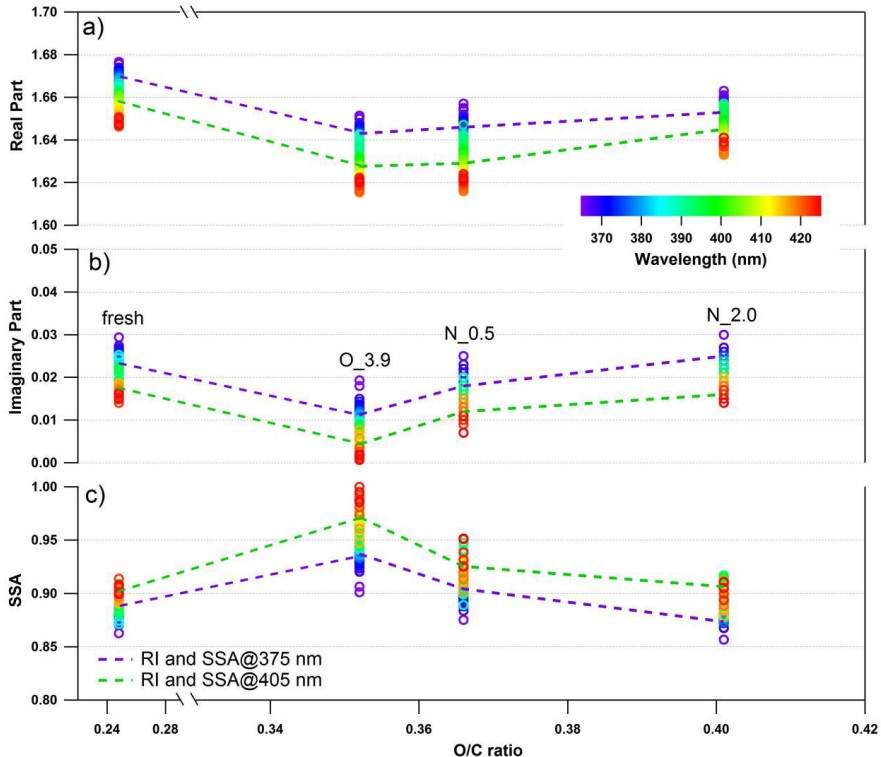

1137

**Figure 9**. Changes of retrieved spectra-dependent RI as a function of O:C ratio for tar ball particles upon NOx-dependent

photochemical oxidation: a) real part, b) imaginary part, and c) SSA calculated from 150 nm particles. For clarity, error bars

for O:C ratio (±0.01), RI (±0.006 for real part, and ±0.003 for imaginary part on average), and SSA (±0.007) are not

shown. O_3.9 represents 3.9 days equivalent atmospheric photochemical aging in absence of NOx, N_0.5 and N_2.0 indicate

photochemical oxidation with 0.5 and 2.0 vol.% N2O addition at ~4.0 days atmospheric oxidation.



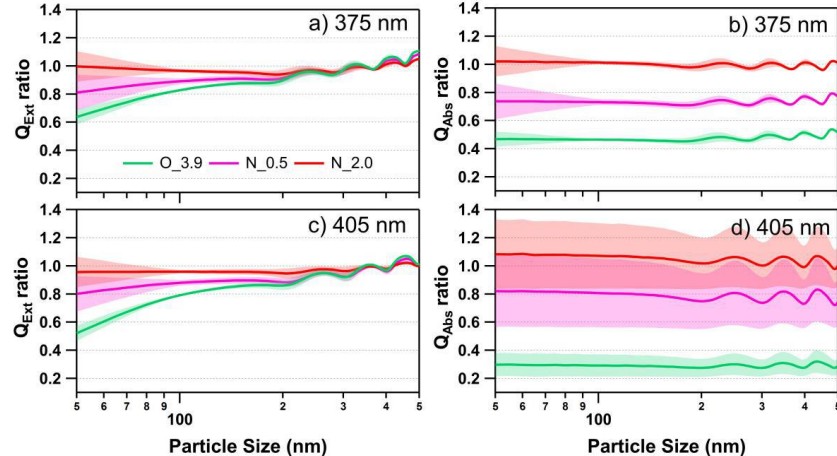


**Figure 10.** Size-resolved light extinction and absorption efficiency ratio of NOx-dependent photooxidized tar balls compared
to the fresh tar ball particles: a) and c) extinction ratios at 375 and 405 nm, b) and d) absorption ratios at 375 and 405 nm.
O_3.9 represents 3.9 days equivalent atmospheric photochemical aging in absence of NOx, N_0.5 and N_2.0 indicate
photochemical oxidation with 0.5 and 2.0 vol.% $N_2O$ addition at ~4.0 days atmospheric oxidation.





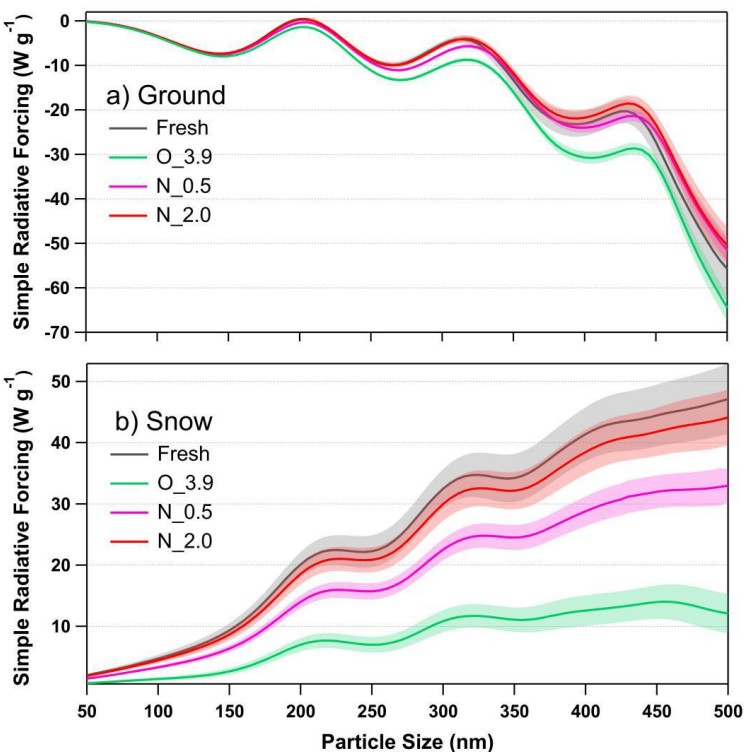

**Figure 11.** Calculated size-resolved simple radiative forcing (SRF, W g$^{-1}$) by tar ball aerosols, integrated over 365~425 nm incident solar irradiation for fresh and NOx-dependent photooxidized tar balls: a) ground-based radiative forcing, b) snow-based radiative forcing.