# Peer review of "Dynamic changes of optical and chemical properties of tar ball"

_Atmospheric Chemistry and Physics, 2018_

## Referee Comment (RC1) · Anonymous Referee #1 · 11 Oct 2018

General This is an interesting paper where 'tar ball aerosol particles' where produced and photochemically aged in an OFR I am not sure whether tar balls in atmospheric aerosol particles are an important topic. As they are mentioned in the title, the reader expects tar balls to be central for this paper, but it seems to me, the real topic of the paper is BrC formed after wood combustion and tar balls have been identified with that BrC. Maybe the authors can straighten this out with regard to the title and the focus of the inroductory text part.

Details

The specific tar ball aerosol generation is intereting.  However, this is a laboratory

method to obtain as much as possible tar balls in the generated aerosols. How realistic is this aerosol for the environment ? I wonder if ome key parameters of the 'tar ball aerosol' generated in the lab should be given in the experimental, or, at the latest, early in the results section: Give the particle size distribution, give a rough chemical composition, OC/EC/ WSOC, weight fraction of tar balls. Doe these particle still contain inorganic constituents ? How much ?

When this is not done, the reader starts into the results section without knowledge about what has actually been generated and is now undergoing heterogeneous oxidation in the OFR.

Much of this information is available, but scattere through the manuscript. I would strongly recommend to introduce a section 'Initial tarball aerosol characteristics after generation'

Maybe the chemical information and the optical measurement results should be separated.

Overall, the manuscript call for a better organisation.

The obtained results are interesting but their atmospheric relevance should be discussed in view of realistic fraction of tar balls over EC or over OC.

Some statements in the conclusionsection are very broad , line 676 ff. Please reconsider.

---

## Referee Comment (RC2) · Anonymous Referee #2 · 15 Oct 2018

This manuscript describes the systematic study of how the chemical and optical properties of laboratory-generated tar ball aerosols change upon reaction with OH, exposure to 254 nm light and oxidation in the presence of NOx. What is really interesting about this study is that the authors extracted the polar (water-soluble) and non-polar fractions and found significant differences in both chemical composition and the responses to oxidation. The formation of organic-nitrates was found to increase absorption, which at least partially offsets the decrease in absorption ("bleaching") observed upon photolysis and oxidation by OH radicals. By using a range of EADs (Equivalent daytime Atmospheric oxidation Days) of  $\sim$  0.5-7 days, the authors explore atmospherically relevant extents of oxidation.
The use and description of a controlled method for obtaining reproducible tar ball extract samples in the laboratory is also important. This procedure allows the authors to make systematic studies of semi-authentic aerosol samples that appear to be consistent with ambient tar ball samples.

Overall, this manuscript is well written, and the data are interpreted carefully. Given the growing evidence of the importance of tar balls in atmospheric aerosols and the open questions about brown carbon's molecular composition, this manuscript is an important contribution in this area and is appropriate for ACPD.

Specific comments:

1. Since the photolysis was carried out at only one wavelength, 254 nm, statements about increases in absorption from the formation of organic-nitrates offsetting bleaching from photolysis should be highly qualified.

2. Equation 4: How is the mass concentration of the solution, C, determined in calculating the MAC (mass absorption cross section)?

3. Line 597: "EAD" should be "EDA".

4. What fraction of the mass and/or the absorption is attributable to the polar vs. the non-polar fractions?

**ACPD**

---

## Author Comment (AC1) · 30 Nov 2018

**atmospheric photochemical aging"** *by Li et al.*

**Li et al.**

yinon.rudich@weizmann.ac.il

**Anonymous Referee #1:** Generally speaking, this is an interesting paper where 'tar ball aerosol particles' were produced and photochemically aged in an OFR. I am not sure whether tar balls in atmospheric aerosol particles are an important topic. As they are mentioned in the title, the reader expects tar balls to be central for this paper, but it seems to me, the real topic of the paper is BrC formed after wood combustion, and tar balls have been identified with that BrC. Maybe the authors can straighten this out with regard to the title and the focus of the introductory text part. The specific tar ball aerosol generation is interesting. However, this is a laboratory method to obtain as much as possible tar balls in the generated aerosols. How realistic is this aerosol for the environment?

I wonder if some key parameters of the 'tar ball aerosol' generated in the lab should be given in the experimental, or, at the latest, early in the results section: Give the particle size distribution, give a rough chemical composition,

OC/EC, WSOC, weight fraction of tar balls. Do these particles still contain inorganic constituents? How much?

When this is not done, the reader starts into the results section without knowledge about what has actually been generated and is now undergoing heterogeneous oxidation in the OFR. Much of this information is available, but scatter through the manuscript. I would strongly recommend to introduce a section 'Initial tar ball aerosol characteristics after generation'. Maybe the chemical information and the optical measurement results should be separated. Overall, the manuscript call for a better organization. The obtained results are interesting but their atmospheric relevance should be discussed in view of realistic fraction of tar balls over EC or over OC. Some statements in the conclusion section are very broad , line 676 ff.

**Authors' reply**: We appreciate the Reviewer's general comments. Tar balls are abundant carbonaceous particles produced by incomplete burning of biofuels. Tar balls contribute a significant number concentration and mass loading in fire emissions. These ubiquitous particles have strong absorption and present potential light absorption across the entire solar spectrum. Due to the strong absorption and high concentrations, they can alter local photochemistry and perturb the radiative balance in the atmosphere on regional and global scales (Pósfai et al., 2004; Hand et al., 2005; Alexander et al., 2008; Chen et al., 2017; Sedlacek III et al., 2018). In ambient biomass burning emissions, tar ball particles coexist with other types of pyrolysis particles (e.g., inorganic salts, fly ashes, amorphous carbonaceous aerosol, soot, etc.) as both internal and external mixtures, and they can undergo rapid atmospheric processes (dilution due to transport, mixing, removal by precipitation, photochemical and nocturnal oxidation, etc.) once they are released from the fires. Therefore, there are many practical difficulties in intensive investigation of the chemical and optical properties of ambient wood tar aerosols.

New methods for laboratory generation of tar ball particles from wood pyrolysis have recently been suggested, and the laboratory-produced tar ball aerosols resemble atmospheric tar balls in most of their observed properties, including the signature feature of spherical morphology and light absorption, elemental ratio, and similar chemical compositions (Tóth et al., 2014, 2018; Hoffer et al., 2016, 2017). These laboratory-generated tar balls are with the similar size range of atmospheric tar ball aerosols (Sedlacek III et al., 2018; Girotto et al., 2018). This opens up the possibility to conduct detailed laboratory experiments on proxies of tar ball aerosols to understand their basic physical and chemical processes. In this study, we followed the reported production protocol (Tóth et al., 2014; Hoffer et al., 2016) to produce liquid wood tar from pyrolysis of wood under high temperature (dry distillation over 500 $^o$C to mimic a smoldering process), then tar ball droplets were generated via aerosolization from these tar solutions. The generated tar droplets underwent subsequent dehydration and heat shock to produce tar ball particles. To study the extensive optical and chemical properties of wood pyrolysis tar materials, polar and nonpolar tar materials were further separated and concentrated to produce polar and nonpolar tar ball particles (Hoffer et al., 2016, 2017).

Before starting the photochemical aging studies, the laboratory-generated fresh tar ball particles were extensively characterized and compared with previous studies on ambient biomass burning organic aerosols (BBOA). As shown in Figure S3 of their morphology (supporting information, SI), the lab-generated tar balls are amorphous, carbonaceous spherules with major carbon and minor oxygen content. The O/C ratio in the bulk tar balls was 0.2~0.4 (Figure 2, Table S1 in SI), which is consistent with ambient tar balls and BBOA (Sumlin et al., 2017, 2018; Aiken et al., 2008; Li et al., 2012; Zhou et al., 2017). The detailed chemical information of fresh tar balls was obtained using high resolution time-of-flight aerosol mass spectrometer (HR-Tof-AMS) and by single-particle laser desorption/resonance enhanced multiphoton ionization mass spectrometry (SP-LD-REMPI-MS). The mass spectra for typical polar and nonpolar tar balls are shown in Figure 2, Figure 3, and Table S2 (SI). It is clear that the tar balls contain significant fractions of aromatic hydrocarbons and methoxy-phenolic compounds. These results are consistent with the finding that BBOA or wood tar aerosols consist of major poly-aromatic hydrocarbons (Tóth et al., 2018, Li et al., 2017; Tivanski et al., 2007; Chen and Bond, 2010). In the manuscript, the mass spectra of these atmospherically-relevant tar balls are extensively described and compared with previous studies on field and domestic released BBOA.

We observed weak signals of inorganic ions (e.g., $NO_3^-$, $Cl^-$, $SO_4^{2-}$, and $NH_4^+$) in the AMS mass spectra of the tar balls, and the mass fractions of these inorganics contributed less than 1.5 wt.% of the bulk aerosols in total (see below Figure R1, pie chart of tar ball chemical composition from AMS measurement), indicating that the generated wood tar particles are dominantly carbonaceous aerosols, the result has been verified by many studies (Pósfai et al., 2004; Hand et al., 2005; Adachi and Buseck, 2011).

[Figure]

Figure R1. Chemical composition of polar and nonpolar tar ball aerosols obtained from AMS measurement

Following the Reviewer's suggestion, we measured the OC-EC-TC fractions (total carbon, TC = OC+EC) in the tar ball aerosols using carbon analyzer based on thermal-optical reflectance (TOR) based on the IMPROVE

protocol. It is clear that the EC content was almost below detection limit for both polar- and nonpolar-tar balls, the slight EC fraction in nonpolar tar ball is less than 0.7% of the TC content and resides in EC1, which can be termed as non-refractory char-EC, empirically defined as EC1－PC. Char-EC is stripped from some OC under oxygen-free heating during OC/EC measurement, which has much weaker absorption, and thus can be classified as brown carbon rather than black carbon (Andreae and Gelencsér, 2006; Arora et al., 2015; Kim et al., 2011;

Han et al., 2008, 2009). Many other studies on biomass burning emissions from wildfires and domestic burning have also reported negligible EC content in tar ball aerosols (Chakrabarty et al., 2010; Tivanski et al., 2007;

Hand et al., 2005; China et al., 2013). These results further support the finding that the composition of the wood tar we generated is dominated by carbonaceous material.

Table R1. Raw blank corrected elemental carbon composition in polar and nonpolar tar ball samples from thermal-optical measurements

| Tar ball  Carbon ($\mu g\ cm^{-2}$) | OC1 | OC2 | OC3 | OC4 | PC | EC1 | EC2 | EC3 | OC | EC |
|---|---|---|---|---|---|---|---|---|---|---|
| Polar | 7.3 | 6.1 | 3.5 | 0.0 | 2.0 | 2.0 | 0.0 | 0.0 | 18.9 | 0.0 |
| Nonpolar | 10.8 | 9.6 | 6.0 | 2.9 | 7.9 | 8.1 | 0.1 | 0.0 | 37.2 | 0.3 |

Note: OC1, OC2, OC3, OC4 are four organic fractions determined at 140, 280, 480, and 580 °C pyrolysis temperatures, respectively, in a helium (He) atmosphere. EC1, EC2, and EC3 are three EC fractions being oxidized at 580, 740, and 840 °C, respectively, in a 2%

$O_2$/98% He atmosphere. OC=OC1+OC2+OC3+OC4+PC, EC=EC1+EC2+EC3-PC.

The light absorption properties of laboratory-generated tar balls were also characterized, see Figure 4 and

Table 2. The wavelength-dependent refractive index (RI) was retrieved over the UV-Vis range (365~425nm) for fresh tar ball aerosols and are reported for the first time and compared with reference values from environmental tar balls and BBOA (Sedlacek III et al., 2018; Sumlin et al., 2017, 2018; Chakrabarty et al., 2010). The identical values at some discrete wavelength and the spectral-dependence trends of the RI further justify the use of the
laboratory-generated tar balls for studying the optical and chemical behavior of ambient tar ball and/or BBOA
in atmospheric process. The RI of tar balls were retrieved based on Mie-Lorenz scattering theory from size- and
wavelength-resolved extinction cross section measurements of 200~350 nm tar ball aerosols (see the method in
manuscript and tar ball size distribution in Figure S1, SI).

This manuscript focuses on the changes in absorption by tar ball aerosols upon photochemical aging. It was
found that OH oxidation bleached the tar balls by depleting of absorbing moieties, while addition of NOx during
aging inhibited the bleaching and even restored the absorption of tar balls by forming nitrogen-containing
functional groups. The changes in the broadband optical properties of the tar balls under different oxidation
conditions are presented in the paper, and the chemical mechanisms attributing to the optical changes were
discussed. The entire manuscript follows the following scheme:

[Figure]

To summarize: this study probed for the first time changes in the optical and chemical properties of proxies
for tar ball aerosols due to OH radical aging with different oxidation extent from 0.7 to 6.7 atmospheric
equivalent days in the presence/absence of NOx. We also assessed pristine photolysis under different photon
fluxes and $O_3$ oxidation of tar balls in the OFR. Finally the atmospheric and climatic implications of tar balls
were discussed using a simple radiative transfer model.

The particulate inorganic and OC-EC composition have been discussed and added in the manuscript and in the
supporting information:

Page 11, Line 272 in manuscript: "as no other refractory elemental carbon (EC) content was detected in our
samples with a thermal-optical analysis method (details see in SI)."

Page 4 in supporting information: Part 2. OC-EC content of fresh polar and nonpolar tar ball aerosols

Page 11, Line 297 in manuscript: "Negligible fractions of inorganics (e.g., sulfate, nitrate, chloride, and ammonium) in tar balls were obtained from AMS measurement as shown in Fig. S2 (SI), and these results further confirm that tar ball aerosols are dominated by carbonaceous compounds with minor amounts of N, S, and Cl (Pósfai et al., 2004; Hand et al., 2005; Adachi and Buseck, 2011). Thereafter, only organics in tar balls are considered,"

Page 5 in supporting information: Part 3.Fresh tar ball composition from HR-Tof-AMS measurement

Reference

Aiken, A. C., Decarlo, P. F., Kroll, J. H., Worsnop, D. R., Huffman, J. A., Docherty, K. S., Ulbrich, I. M., Mohr, C., Kimmel, J. R., and Sueper, D.: O/C and OM/OC ratios of primary, secondary, and ambient organic aerosols with high-resolution time-of-flight aerosol mass spectrometry, Environ. Sci. Technol., 42, 4478-4485, doi: 10.1021/es703009q, 2008.

Alexander, D. T., Crozier, P. A., and Anderson, J. R.: Brown carbon spheres in East Asian outflow and their optical properties, Science, 321, 833-836, doi: 10.1126/science.1155296, 2008.

Andreae, M., and Gelencsér, A.: Black carbon or brown carbon? The nature of light-absorbing carbonaceous aerosols, Atmos. Chem. Phys., 6, 3131-3148, doi:10.5194/acp-6-3131-2006, 2006.

Arora, P., and Jain, S.: Estimation of organic and elemental carbon emitted from wood burning in traditional and improved cookstoves using controlled cooking test, Environ. Sci. Technol., 49, 3958-3965, doi:10.1021/es504012v, 2015.

Asa-Awuku, A., Sullivan, A., Hennigan, C., Weber, R., and Nenes, A.: Investigation of molar volume and surfactant characteristics of water-soluble organic compounds in biomass burning aerosol, Atmos. Chem. Phys., 8, 799-812, doi:10.5194/acp-8-799-2008, 2008.

Bluvshtein, N.; Lin, P.; Flores, J. M.; Segev, L.; Mazar, Y.; Tas, E.; Snider, G.; Weagle, C.; Brown, S. S.; Laskin, A., Broadband optical properties of biomass‐burning aerosol and identification of brown carbon chromophores. J. Geophys. Res. Atmos., 122, 5441-5456, doi:10.1002/2016JD026230, 2017.

Chakrabarty, R., Moosmüller, H., Chen, L.-W., Lewis, K., Arnott, W., Mazzoleni, C., Dubey, M., Wold, C., Hao, W., and Kreidenweis, S.: Brown carbon in tar balls from smoldering biomass combustion, Atmos. Chem. Phys., 10, 6363-6370, doi:10.5194/acp-10-6363-2010, 2010.

Chen, J., Li, C., Ristovski, Z., Milic, A., Gu, Y., Islam, M. S., Wang, S., Hao, J., Zhang, H., and He, C.: A review of biomass burning: Emissions and impacts on air quality, health and climate in China, Sci. Total Environ., 579, 1000-1034, doi:10.1016/j.scitotenv.2016.11.025, 2017.

Chen, Y., and Bond, T.: Light absorption by organic carbon from wood combustion, Atmos. Chem. Phys., 10, 1773-1787, doi:10.5194/acp-10-1773-2010, 2010.

Girotto, G., China, S., Bhandari, J., Gorkowski, K., Scarnato, B., Capek, T., Marinoni, A., Veghte, D., Kulkarni, G., and Aiken, A.: Fractal-like Tar Ball Aggregates from Wildfire Smoke, Environ. Sci. Technol. Lett., doi:10.1021/acs.estlett.8b00229, 2018.

Han, Y., Han, Z., Cao, J., Chow, J., Watson, J., An, Z., Liu, S., and Zhang, R.: Distribution and origin of carbonaceous aerosol over a rural high-mountain lake area, Northern China and its transport significance, Atmos. Environ., 42, 2405-2414, doi:10.1016/j.atmosenv.2007.12.020, 2008.

Han, Y., Lee, S., Cao, J., Ho, K., and An, Z.: Spatial distribution and seasonal variation of char-EC and soot-EC in the atmosphere over China, Atmos. Environ., 43, 6066-6073, doi:10.1016/j.atmosenv.2009.08.018, 2009.

Han, Y., Cao, J., Lee, S., Ho, K., and An, Z.: Different characteristics of char and soot in the atmosphere and their ratio as an indicator for source identification in Xi'an, China, Atmos. Chem. Phys., 10, 595-607, doi:10.5194/acp-10-595-2010, 2010.

Hand, J. L., Malm, W., Laskin, A., Day, D., Lee, T.b, Wang, C., Carrico, C., Carrillo, J., Cowin, J. P., and Collett, J.: Optical, physical, and chemical properties of tar balls observed during the Yosemite Aerosol Characterization Study, J. Geophys. Res. Atmos., 110, doi:10.1029/2004JD005728, 2005.

Hoffer, A., Tóth, A., Nyirő-Kósa, I., Pósfai, M., and Gelencsér, A.: Light absorption properties of laboratory-generated tar ball particles, Atmos. Chem. Phys., 16, 239-246, doi:10.5194/acp-16-239-2016, 2016.

Hoffer, A., Tóth, Á., Pósfai, M., Chung, C. E., and Gelencsér, A.: Brown carbon absorption in the red and near-infrared spectral region, Atmos. Chem. Phys., 10, 2353-2359, doi:10.5194/acp-2016-452, 2017.

Kim, K. H., Sekiguchi, K., Kudo, S., and Sakamoto, K.: Characteristics of Atmospheric Elemental Carbon(Char and Soot) in Ultrafine and Fine Particles in a Roadside Environment, Japan, Aerosol Air Qual. Res., 11, 1-12, doi:10.4209/aaqr.2010.07.0061, 2011.

Li, C., Hu, Y., Zhang, F., Chen, J., Ma, Z., Ye, X., Yang, X., Wang, L., Tang, X., and Zhang, R.: Multi-pollutant emissions from the burning of major agricultural residues in China and the related health-economic effects, Atmos. Chem. Phys., 17, 4957-4988, doi: 10.5194/acp-17-4957-2017, 2017.

Li, Y. J., Yeung, J. W., Leung, T. P., Lau, A. P., and Chan, C. K.: Characterization of organic particles from incense burning using an aerodyne high-resolution time-of-flight aerosol mass spectrometer, Aerosol Sci. Technol., 46, 654-665, doi:10.1080/02786826.2011.653017, 2012.

Lin, P., Bluvshtein, N., Rudich, Y., Nizkorodov, S. A., Laskin, J., and Laskin, A.: Molecular Chemistry of Atmospheric Brown Carbon Inferred from a Nationwide Biomass Burning Event, Environ. Sci. Technol., 51, 11561-11570, doi:10.1021/acs.est.7b02276, 2017.

Lin, P.; Fleming, L. T.; Nizkorodov, S. A.; Laskin, J.; Laskin, A., Comprehensive Molecular Characterization of Atmospheric Brown Carbon by High Resolution Mass Spectrometry with Electrospray and Atmospheric Pressure Photoionization. Anal. Chem., 90, 12439-12502, doi:10.1021/acs.analchem.8b02177, 2018.

Pósfai, M., Gelencsér, A., Simonics, R., Arató, K., Li, J., Hobbs, P. V., and Buseck, P. R.: Atmospheric tar balls: Particles from biomass and biofuel burning, J. Geophys. Res. Atmos., 109, doi:10.1029/2003JD004169, 2004.

Rajput, P., and Sarin, M.: Polar and non-polar organic aerosols from large-scale agricultural-waste burning emissions in Northern India: implications to organic mass-to-organic carbon ratio, Chemosphere, 103, 74-79, doi:10.1016/j.chemosphere.2013.11.028, 2014.

Sedlacek III, A. J., Buseck, P. R., Adachi, K., Onasch, T. B., Springston, S. R., and Kleinman, L.: Formation and evolution of Tar Balls from Northwestern US wildfires, Atmos. Chem. Phys., 18, 11289-11301, doi:10.5194/acp-18-11289-2018, 2018.

Sengupta, D., Samburova, V, Bhattarai, C., Kirillova, E., Mazzoleni, L., Iaukea-Lum, M., Watts, A., Moosmüller, H., and Khlystov, A.: Light absorption by polar and non-polar aerosol compounds from laboratory biomass combustion, Atmos. Chem. Phys., 18, 10849-10867, doi:10.5194/acp-18-10849-2018, 2018.

Sumlin, B. J., Pandey, A., Walker, M. J., Pattison, R. S., Williams, B. J., and Chakrabarty, R. K.: Atmospheric Photooxidation Diminishes Light Absorption by Primary Brown Carbon Aerosol from Biomass Burning, Environ. Sci. Technol. Lett., 4, 540-545, doi:10.1021/acs.estlett.7b00393, 2017.

Sumlin, B. J., Oxford, C. R., Seo, B., Pattison, R. R., Williams, B. J., and Chakrabarty, R. K.: Density and homogeneous internal composition of primary brown carbon aerosol, Environ. Sci. Technol., 52, 3982-3989, doi: 10.1021/acs.est.8b00093, 2018.

Tivanski, A. V., Hopkins, R. J., Tyliszczak, T., and Gilles, M. K.: Oxygenated interface on biomass burn tar balls determined by single particle scanning transmission X-ray microscopy, J. Phys. Chem. A, 111, 5448-5458, doi: 10.1021/jp070155u, 2007.

Tóth, A., Hoffer, A., Nyirő-Kósa, I., Pósfai, M., and Gelencsér, A.: Atmospheric tar balls: aged primary droplets from biomass burning?, Atmos. Chem. Phys., 14, 6669-6675, doi:10.5194/acp-14-6669-2014, 2014.

Tóth, Á., Hoffer, A., Pósfai, M., Ajtai, T., Kónya, Z., Blazsó, M., Czégény, Z., Kiss, G., Bozóki, Z., and Gelencsér, A.: Chemical characterization of laboratory-generated tar ball particles, Atmos. Chem. Phys., 18, 10407-10418, doi:10.5194/acp-18-10407-2018, 2018.

Zhou, S., Collier, S., Jaffe, D. A., Briggs, N. L., Hee, J., Sedlacek III, A. J., Kleinman, L., Onasch, T. B., and Zhang, Q.: Regional influence of wildfires on aerosol chemistry in the western US and insights into atmospheric aging of biomass burning organic aerosol, Atmos. Chem. Phys., 17, 2477-2493, doi:10.5194/acp-17-2477-2017, 2017.

---

## Author Comment (AC2) · 30 Nov 2018

**Anonymous Referee #2:** This manuscript describes the systematic study of how the chemical and optical properties of laboratory-generated tar ball aerosols change upon reaction with OH, exposure to 254 nm light and oxidation in the presence of NOx. What is really interesting about this study is that the authors extracted the polar (water-soluble) and non-polar fractions and found significant differences in both chemical composition and the responses to oxidation. The formation of organic-nitrates was found to increase absorption, which at least partially offsets the decrease in absorption ("bleaching") observed upon photolysis and oxidation by OH radicals. By using a range of EADs (Equivalent daytime Atmospheric oxidation Days) of ~0.5-7 days, the authors explore atmospherically relevant extents of oxidation. The use and description of a controlled method for obtaining reproducible tar ball extract samples in the laboratory is also important. This procedure allows the authors to make systematic studies of semi-authentic aerosol samples that appear to be consistent with ambient tar ball samples. Overall, this manuscript is well written, and the data are interpreted carefully. Given the growing evidence of the importance of tar balls in atmospheric aerosols and the open questions about brown carbon's molecular composition, this manuscript is an important contribution in this area and is appropriate for ACPD.

Specific comments:

1. Since the photolysis was carried out at only one wavelength, 254 nm, statements about increases in absorption from the formation of organic-nitrates offsetting bleaching from photolysis should be highly qualified.

**Authors' reply**: We thanks the Reviewer for the thoughtful comments. Tar ball particles underwent photochemical aging in the OFR. This included NOx-dependent OH photooxidation and photolysis at 254 nm. We suggest that NOx-dependent OH radical aging should be the dominant photochemical pathway that bleach, darken and oxidize tar balls (Sumlin et al., 2017). The role of photolysis in tar ball chemical and optical changes were investigated, and the results are discussed in detail in the Supporting Information Part 12 (**Optical and chemical changes for tar ball aerosols due to photolysis from UV light irradiation in the OFR**). Specifically, we exposed the tar balls to the same aging condition without $O_3$ and NOx addition, where the actual UV flux is higher without $O_3$, $RO_2/HO_2$ radicals and NOx that absorb some of the photons. However, no significant chemical and RI changes were observed in the tar balls even under maximum UV flux in the OFR. On the experimental time scale (residence time ~144s in the OFR), the small observed changes during photolysis can be neglected compared to the extensive bleaching and oxidation observed under the same conditions in presence of OH radical. We have clarified these points in the paper (Page 18, Line 496-504).

It was also found that addition of NOx restored absorption and counteracted the OH photochemical bleaching of the tar balls to facilitate absorption under high NOx concentration. We suggest that the formation of organic nitrates offsets the bleaching from photolysis and from OH photooxidation to eventually enhance light absorption of the tar balls. It is correct that 254nm UV photolysis in the OFR does not represent the solar irradiation in the atmosphere, it is arbitrary to state that increases in absorption from the formation of organic-nitrates offset bleaching from photolysis. We corrected this statement (Page 2, Line 35-36), and we will quantify the photolysis and OH radical oxidation effect in a following study, and study the chemical process of some related organic-nitrates during their lifetime in
the atmosphere.

Page 2, Line 35-36 in manuscript: "These observations suggest that secondary organic nitrate
formation counteracts the bleaching by OH radical photooxidation to eventually regain some
absorption of the tar balls aerosols."

2. Equation 4: How is the mass concentration of the solution, C, determined in calculating the
MAC (mass absorption cross section)?

**Authors' reply**: The mass concentration of BrC extracted in methanol solution was calculated
to be in the range of 0.01~0.02 g $L^{-1}$. The BrC concentration was calculated from filter mass
loading of tar ball aerosol (100~150 μg), methanol volume (7.5 mL) assuming hundred
percent extraction efficiency, and OM/OC ratio from AMS measurement. The mass loading
for each filter sample was estimated from the aerosol volume distribution measured by SMPS,
particle density measured from AAC-SMPS (~1.24-1.33 g $cm^{-3}$), and the total sampling
volume. The BrC mass concentration can be calculated by the function shown below:

$$C_{BrC} = \frac{V_{tar} \times \rho_{eff} \times t \times v}{V_{methanol}} \times r$$

Where $V_{tar}$ is the tar ball aerosol volume concentration measured by SMPS between 14 and
670 nm, $\rho_{eff}$ is effective density calculated from aerodynamic and electric-mobility
distribution for fixed monodispersed tar ball, $t$ is the filter sampling time, $v$ is filter sampling
flow, $V_{methanol}$ is the total volume of solvent used to extract the filter sample, $r$ is organic
carbon (OC) fraction in tar ball aerosol, which can be derived from AMS measurement of
OC/OM with the assumption that the EC content can be neglected in tar ball aerosol (as
discussed above).

We added this information in the methods part of the manuscript:

Page10 Line 270 in manuscript: "$C$ is the extracted organic carbon mass concentration in
solvent (g $m^{-3}$), which can be determined directly by normalizing the extract concentration
and OC mass fraction for tar balls as OC/OM obtained from AMS data, as no other refractory
elemental carbon (EC) content was detected in our samples (details see in SI)."

3. Line 597: "EAD" should be "EDA".

**Authors' reply**: Thanks. "EAD" has been corrected in Page 22, Lin 620.

4. What fraction of the mass and/or the absorption is attributable to the polar vs. the non-polar
fractions?

**Authors' reply**: In this study, light absorbing properties of polar and nonpolar chemical
matrix from wood pyrolysis were characterized. We found that the refractive index for the
mixture of polar and nonpolar organic fractions fit both volume and molar mixing rules (see
supporting information of prediction of mixture tar ball optical properties based on different
mixing rules). However, the actual fractions of the polar and nonpolar compounds contributing to the mass/absorption of ambient tar ball or BBOA are still under study, as they
vary with biofuel, burning conditions, atmospheric processes, and also the method/efficiency
used to classify the polar and nonpolar materials from the sample (Sengupta., et al., 2018; Lin
et al., 2017; Chen and Bond, et al., 2010; Sumlin et al., 2017) (Page 6, Line 133-137).
According to most previous studies, the nonpolar or less polar fractions have higher
absorption properties compared with the polar fraction from BBOA. Sengupta et al. (2018)
reported that the fuel mass normalized nonpolar fraction is 2-3 times more absorbing than the
polar fraction, and complex fuel-dependence and atmospheric aging dependence were
observed for absorption changes of both fractions. Lin et al. (2017) and Bluvshtein et al.
(2017) tracked the absorption changes of BBOA with respect to different solvent extraction
during a whole fire episode, and found distinct different absorbing features for the
water-extracted fraction compared to the organic solvent extracts, while acetonitrile and
orgmix (acetonitrile: dichloromethane: hexane = 2:2:1, $v/v$) extracts exhibited similar
absorption. According to Lin et al. (2017), from the beginning to the end of the fire event,
organic extracts showed higher light absorption at shorter wavelength ($\lambda$<380 nm). In contrast,
higher light absorption with an absorption feature around 450 nm (attributed to nitro-phenols)
hump was observed for water extracts.

Rajput et al. (2014) classified the mass fractions of polar and nonpolar organic matter from
agricultural-waste burning emissions. Nearly 85 wt.% of the burning organic emissions was
attributed to a polar fraction, and this fraction from wheat residue burning was much lower
than from paddy burning. Asa-Awuku et al. (2008) suggested that relative molar proportion of
nonpolar (hydrophobic) to polar (hydrophilic) compounds in original biomass burning aerosol
is 1:3 from Köhler Theory Analysis (Page 6, Line 133-137; Page 17, Line 451-456).

In short, the polar fraction dominates the bulk organic aerosol from biomass burning.
However, the real polar fraction in the tar balls is undefined, as tar balls belong to BBOA but
from specific burning condition with special physicochemical properties. The topic raised in
the Reviewer's question will be part of our following research topic to investigate the exact
contribution of both the polar/nonpolar matrix and to identify specific chromophore
molecules to tar ball aerosol absorption/mass. We have sent the samples of the fresh and aged
laboratory-generated polar/nonpolar tar ball for extensive molecular chemical analysis using
HPLC/PDA/HRMS (high performance liquid chromatography platform coupled to
photo-diode array and high resolution mass spectrometry detectors) and other common MS
techniques (e.g., ESI/APPI-MS, electrospray ionization/atmospheric pressure photo ionization
mass spectrometry), the results will be published in a different manuscript.

We have added the discussion of polar and nonpolar fractions contribution to tar ball
aerosols mass and absorption in the manuscript:

Page 6, Line 133-137: "The actual fractions of the polar and nonpolar compounds
contributing to the mass of ambient tar ball or biomass burning organic aerosol (BBOA) can
vary with biofuel sources, burning condition, atmospheric process, and also method/efficiency
to classify the polar and nonpolar materials from the sample (Sengupta., et al., 2018; Lin et al.,
2017, 2018; Chen and Bond, et al., 2010; Rajput et al., 2008),"

Page 17, Line 451-456: "As mentioned above, the real fractions of polar and nonpolar BrC
contributing to the mass/absorption of BBOA are undefined, some investigations report that the polar BrC dominate the tar balls' mass (50~85%), but contribute less to the absorption in

BBOA (less than 40%) (Asa-Awuku et al., 2008; Bluvshtein et al., 2017; Lin et al., 2017,

2018; Rajput et al., 2014; Sengupta et al., 2018). The "linear mixing rule" confirmed in this study should be helpful in the mathematical modeling to assess the climatic impacts of biomass burning related BrC aerosol, when their chemical composition is classified."

Reference

Asa-Awuku, A., Sullivan, A., Hennigan, C., Weber, R., and Nenes, A.: Investigation of molar volume and surfactant characteristics of water-soluble organic compounds in biomass burning aerosol, Atmos. Chem. Phys., 8, 799-812, doi:10.5194/acp-8-799-2008, 2008.

Bluvshtein, N.; Lin, P.; Flores, J. M.; Segev, L.; Mazar, Y.; Tas, E.; Snider, G.; Weagle, C.; Brown, S. S.; Laskin, A., Broadband optical properties of biomass‐burning aerosol and identification of brown carbon chromophores. J. Geophys. Res. Atmos., 122, 5441-5456, doi:10.1002/2016JD026230, 2017.

Chen, Y., and Bond, T.: Light absorption by organic carbon from wood combustion, Atmos. Chem. Phys., 10, 1773-1787, doi:10.5194/acp-10-1773-2010, 2010.

Lin, P., Bluvshtein, N., Rudich, Y., Nizkorodov, S. A., Laskin, J., and Laskin, A.: Molecular Chemistry of Atmospheric Brown Carbon Inferred from a Nationwide Biomass Burning Event, Environ. Sci. Technol., 51, 11561-11570, doi:10.1021/acs.est.7b02276, 2017.

Lin, P.; Fleming, L. T.; Nizkorodov, S. A.; Laskin, J.; Laskin, A., Comprehensive Molecular Characterization of Atmospheric Brown Carbon by High Resolution Mass Spectrometry with Electrospray and Atmospheric Pressure Photoionization. Anal. Chem., 90, 12439-12502, doi:10.1021/acs.analchem.8b02177, 2018.

Rajput, P., and Sarin, M.: Polar and non-polar organic aerosols from large-scale agricultural-waste burning emissions in Northern India: implications to organic mass-to-organic carbon ratio, Chemosphere, 103, 74-79, doi:10.1016/j.chemosphere.2013.11.028, 2014.

Sengupta, D., Samburova, V., Bhattarai, C., Kirillova, E., Mazzoleni, L., Iaukea-Lum, M., Watts, A., Moosmüller, H., and Khlystov, A.: Light absorption by polar and non-polar aerosol compounds from laboratory biomass combustion, Atmos. Chem. Phys., 18, 10849-10867, doi:10.5194/acp-18-10849-2018, 2018.

Sumlin, B. J., Pandey, A., Walker, M. J., Pattison, R. S., Williams, B. J., and Chakrabarty, R. K.: Atmospheric Photooxidation Diminishes Light Absorption by Primary Brown Carbon Aerosol from Biomass Burning, Environ. Sci. Technol. Lett., 4, 540-545, doi:10.1021/acs.estlett.7b00393, 2017.

---

## Author Response (AR1)

**atmospheric photochemical aging**" *by Li et al.*

**Li et al.**

yinon.rudich@weizmann.ac.il

**Anonymous Referee #1:** Generally speaking, this is an interesting paper where 'tar ball aerosol particles' were produced and photochemically aged in an OFR. I am not sure whether tar balls in atmospheric aerosol particles are an important topic. As they are mentioned in the title, the reader expects tar balls to be central for this paper, but it seems to me, the real topic of the paper is BrC formed after wood combustion, and tar balls have been identified with that BrC. Maybe the authors can straighten this out with regard to the title and the focus of the introductory text part. The specific tar ball aerosol generation is interesting. However, this is a laboratory method to obtain as much as possible tar balls in the generated aerosols. How realistic is this aerosol for the environment?

I wonder if some key parameters of the 'tar ball aerosol' generated in the lab should be given in the experimental, or, at the latest, early in the results section: Give the particle size distribution, give a rough chemical composition,

OC/EC, WSOC, weight fraction of tar balls. Do these particles still contain inorganic constituents? How much?

When this is not done, the reader starts into the results section without knowledge about what has actually been generated and is now undergoing heterogeneous oxidation in the OFR. Much of this information is available, but scatter through the manuscript. I would strongly recommend to introduce a section 'Initial tar ball aerosol characteristics after generation'. Maybe the chemical information and the optical measurement results should be separated. Overall, the manuscript call for a better organization. The obtained results are interesting but their atmospheric relevance should be discussed in view of realistic fraction of tar balls over EC or over OC. Some statements in the conclusion section are very broad , line 676 ff.

**Authors' reply**: We appreciate the Reviewer's general comments. Tar balls are abundant carbonaceous particles produced by incomplete burning of biofuels. Tar balls contribute a significant number concentration and mass loading in fire emissions. These ubiquitous particles have strong absorption and present potential light absorption across the entire solar spectrum. Due to the strong absorption and high concentrations, they can alter local photochemistry and perturb the radiative balance in the atmosphere on regional and global scales (Pósfai et al., 2004; Hand et al., 2005; Alexander et al., 2008; Chen et al., 2017; Sedlacek III et al., 2018). In ambient biomass burning emissions, tar ball particles coexist with other types of pyrolysis particles (e.g., inorganic salts, fly ashes, amorphous carbonaceous aerosol, soot, etc.) as both internal and external mixtures, and they can undergo rapid atmospheric processes (dilution due to transport, mixing, removal by precipitation, photochemical and nocturnal oxidation, etc.) once they are released from the fires. Therefore, there are many practical difficulties in intensive investigation of the chemical and optical properties of ambient wood tar aerosols.

New methods for laboratory generation of tar ball particles from wood pyrolysis have recently been suggested, and the laboratory-produced tar ball aerosols resemble atmospheric tar balls in most of their observed properties, including the signature feature of spherical morphology and light absorption, elemental ratio, and similar chemical compositions (Tóth et al., 2014, 2018; Hoffer et al., 2016, 2017). These laboratory-generated tar balls are with the similar size range of atmospheric tar ball aerosols (Sedlacek III et al., 2018; Girotto et al., 2018). This opens up the possibility to conduct detailed laboratory experiments on proxies of tar ball aerosols to understand their basic physical and chemical processes. In this study, we followed the reported production protocol (Tóth et al., 2014; Hoffer et al., 2016) to produce liquid wood tar from pyrolysis of wood under high temperature (dry distillation over 500 $^o$C to mimic a smoldering process), then tar ball droplets were generated via aerosolization from these tar solutions. The generated tar droplets underwent subsequent dehydration and heat shock to produce tar ball particles. To study the extensive optical and chemical properties of wood pyrolysis tar materials, polar and nonpolar tar materials were further separated and concentrated to produce polar and nonpolar tar ball particles (Hoffer et al., 2016, 2017).

Before starting the photochemical aging studies, the laboratory-generated fresh tar ball particles were extensively characterized and compared with previous studies on ambient biomass burning organic aerosols (BBOA). As shown in Figure S3 of their morphology (supporting information, SI), the lab-generated tar balls are amorphous, carbonaceous spherules with major carbon and minor oxygen content. The O/C ratio in the bulk tar balls was 0.2~0.4 (Figure 2, Table S1 in SI), which is consistent with ambient tar balls and BBOA (Sumlin et al., 2017, 2018; Aiken et al., 2008; Li et al., 2012; Zhou et al., 2017). The detailed chemical information of fresh tar balls was obtained using high resolution time-of-flight aerosol mass spectrometer (HR-Tof-AMS) and by single-particle laser desorption/resonance enhanced multiphoton ionization mass spectrometry (SP-LD-REMPI-MS). The mass spectra for typical polar and nonpolar tar balls are shown in Figure 2, Figure 3, and Table S2 (SI). It is clear that the tar balls contain significant fractions of aromatic hydrocarbons and methoxy-phenolic compounds. These results are consistent with the finding that BBOA or wood tar aerosols consist of major poly-aromatic hydrocarbons (Tóth et al., 2018, Li et al., 2017; Tivanski et al., 2007; Chen and Bond, 2010). In the manuscript, the mass spectra of these atmospherically-relevant tar balls are extensively described and compared with previous studies on field and domestic released BBOA.

We observed weak signals of inorganic ions (e.g., $NO_3^-$, $Cl^-$, $SO_4^{2-}$, and $NH_4^+$) in the AMS mass spectra of the tar balls, and the mass fractions of these inorganics contributed less than 1.5 wt.% of the bulk aerosols in total (see below Figure R1, pie chart of tar ball chemical composition from AMS measurement), indicating that the generated wood tar particles are dominantly carbonaceous aerosols, the result has been verified by many studies (Pósfai et al., 2004; Hand et al., 2005; Adachi and Buseck, 2011).

[Figure]

Figure R1. Chemical composition of polar and nonpolar tar ball aerosols obtained from AMS measurement

Following the Reviewer's suggestion, we measured the OC-EC-TC fractions (total carbon, TC = OC+EC) in the tar ball aerosols using carbon analyzer based on thermal-optical reflectance (TOR) based on the IMPROVE protocol. It is clear that the EC content was almost below detection limit for both polar- and nonpolar-tar balls, the slight EC fraction in nonpolar tar ball is less than 0.7% of the TC content and resides in EC1, which can be termed as non-refractory char-EC, empirically defined as EC1－PC. Char-EC is stripped from some OC under oxygen-free heating during OC/EC measurement, which has much weaker absorption, and thus can be classified as brown carbon rather than black carbon (Andreae and Gelencsér, 2006; Arora et al., 2015; Kim et al., 2011; Han et al., 2008, 2009). Many other studies on biomass burning emissions from wildfires and domestic burning have also reported negligible EC content in tar ball aerosols (Chakrabarty et al., 2010; Tivanski et al., 2007; Hand et al., 2005; China et al., 2013). These results further support the finding that the composition of the wood tar we generated is dominated by carbonaceous material.

Table R1. Raw blank corrected elemental carbon composition in polar and nonpolar tar ball samples from thermal-optical measurements

| Tar ball  Carbon ($\mu g\ cm^{-2}$) | OC1 | OC2 | OC3 | OC4 | PC | EC1 | EC2 | EC3 | OC | EC |
|---|---|---|---|---|---|---|---|---|---|---|
| Polar | 7.3 | 6.1 | 3.5 | 0.0 | 2.0 | 2.0 | 0.0 | 0.0 | 18.9 | 0.0 |
| Nonpolar | 10.8 | 9.6 | 6.0 | 2.9 | 7.9 | 8.1 | 0.1 | 0.0 | 37.2 | 0.3 |

Note: OC1, OC2, OC3, OC4 are four organic fractions determined at 140, 280, 480, and 580 $^{o}C$ pyrolysis temperatures, respectively, in a helium (He) atmosphere. EC1, EC2, and EC3 are three EC fractions being oxidized at 580, 740, and 840 $^{o}C$, respectively, in a 2% $O_2$/98% He atmosphere. OC=OC1+OC2+OC3+OC4+PC, EC=EC1+EC2+EC3-PC.

The light absorption properties of laboratory-generated tar balls were also characterized, see Figure 4 and Table 2. The wavelength-dependent refractive index (RI) was retrieved over the UV-Vis range (365~425nm) for fresh tar ball aerosols and are reported for the first time and compared with reference values from environmental tar balls and BBOA (Sedlacek III et al., 2018; Sumlin et al., 2017, 2018; Chakrabarty et al., 2010). The identical values at some discrete wavelength and the spectral-dependence trends of the RI further justify the use of the
laboratory-generated tar balls for studying the optical and chemical behavior of ambient tar ball and/or BBOA
in atmospheric process. The RI of tar balls were retrieved based on Mie-Lorenz scattering theory from size- and
wavelength-resolved extinction cross section measurements of 200~350 nm tar ball aerosols (see the method in
manuscript and tar ball size distribution in Figure S1, SI).

This manuscript focuses on the changes in absorption by tar ball aerosols upon photochemical aging. It was
found that OH oxidation bleached the tar balls by depleting of absorbing moieties, while addition of NOx during
aging inhibited the bleaching and even restored the absorption of tar balls by forming nitrogen-containing
functional groups. The changes in the broadband optical properties of the tar balls under different oxidation
conditions are presented in the paper, and the chemical mechanisms attributing to the optical changes were
discussed. The entire manuscript follows the following scheme:

[Figure]

To summarize: this study probed for the first time changes in the optical and chemical properties of proxies
for tar ball aerosols due to OH radical aging with different oxidation extent from 0.7 to 6.7 atmospheric
equivalent days in the presence/absence of NOx. We also assessed pristine photolysis under different photon
fluxes and $O_3$ oxidation of tar balls in the OFR. Finally the atmospheric and climatic implications of tar balls
were discussed using a simple radiative transfer model.

The particulate inorganic and OC-EC composition have been discussed and added in the manuscript and in the
supporting information:

Page 11, Line 272 in manuscript: "as no other refractory elemental carbon (EC) content was detected in our
samples with a thermal-optical analysis method (details see in SI)."

Page 4 in supporting information: Part 2. OC-EC content of fresh polar and nonpolar tar ball aerosols

Page 11, Line 297 in manuscript: "Negligible fractions of inorganics (e.g., sulfate, nitrate, chloride, and ammonium) in tar balls were obtained from AMS measurement as shown in Fig. S2 (SI), and these results further confirm that tar ball aerosols are dominated by carbonaceous compounds with minor amounts of N, S, and Cl (Pósfai et al., 2004; Hand et al., 2005; Adachi and Buseck, 2011). Thereafter, only organics in tar balls are considered,"

Page 5 in supporting information: Part 3.Fresh tar ball composition from HR-Tof-AMS measurement

volume. The BrC mass concentration can be calculated by the function shown below:

$$C_{BrC} = \frac{V_{tar} \times \rho_{eff} \times t \times v}{V_{methanol}} \times r$$

Where $V_{tar}$ is the tar ball aerosol volume concentration measured by SMPS between 14 and
670 nm, $\rho_{eff}$ is effective density calculated from aerodynamic and electric-mobility
distribution for fixed monodispersed tar ball, $t$ is the filter sampling time, $v$ is filter sampling
flow, $V_{methanol}$ is the total volume of solvent used to extract the filter sample, $r$ is organic
carbon (OC) fraction in tar ball aerosol, which can be derived from AMS measurement of
OC/OM with the assumption that the EC content can be neglected in tar ball aerosol (as
discussed above).

We added this information in the methods part of the manuscript:

Page10 Line 270 in manuscript: "$C$ is the extracted organic carbon mass concentration in
solvent (g m$^{-3}$), which can be determined directly by normalizing the extract concentration
and OC mass fraction for tar balls as OC/OM obtained from AMS data, as no other refractory
elemental carbon (EC) content was detected in our samples (details see in SI)."

3. Line 597: "EAD" should be "EDA".

**Authors' reply**: Thanks. "EAD" has been corrected in Page 22, Lin 620.

4. What fraction of the mass and/or the absorption is attributable to the polar vs. the non-polar
fractions?

**Authors' reply**: In this study, light absorbing properties of polar and nonpolar chemical
matrix from wood pyrolysis were characterized. We found that the refractive index for the
mixture of polar and nonpolar organic fractions fit both volume and molar mixing rules (see
supporting information of prediction of mixture tar ball optical properties based on different
mixing rules). However, the actual fractions of the polar and nonpolar compounds contributing to the mass/absorption of ambient tar ball or BBOA are still under study, as they
vary with biofuel, burning conditions, atmospheric processes, and also the method/efficiency
used to classify the polar and nonpolar materials from the sample (Sengupta., et al., 2018; Lin
et al., 2017; Chen and Bond, et al., 2010; Sumlin et al., 2017) (Page 6, Line 133-137).
According to most previous studies, the nonpolar or less polar fractions have higher
absorption properties compared with the polar fraction from BBOA. Sengupta et al. (2018)
reported that the fuel mass normalized nonpolar fraction is 2-3 times more absorbing than the
polar fraction, and complex fuel-dependence and atmospheric aging dependence were
observed for absorption changes of both fractions. Lin et al. (2017) and Bluvshtein et al.
(2017) tracked the absorption changes of BBOA with respect to different solvent extraction
during a whole fire episode, and found distinct different absorbing features for the
water-extracted fraction compared to the organic solvent extracts, while acetonitrile and
orgmix (acetonitrile: dichloromethane: hexane = 2:2:1, *v/v*) extracts exhibited similar
absorption. According to Lin et al. (2017), from the beginning to the end of the fire event,
organic extracts showed higher light absorption at shorter wavelength ($\lambda<380$ nm). In contrast,
higher light absorption with an absorption feature around 450 nm (attributed to nitro-phenols)
hump was observed for water extracts.

Rajput et al. (2014) classified the mass fractions of polar and nonpolar organic matter from
agricultural-waste burning emissions. Nearly 85 wt.% of the burning organic emissions was
attributed to a polar fraction, and this fraction from wheat residue burning was much lower
than from paddy burning. Asa-Awuku et al. (2008) suggested that relative molar proportion of
nonpolar (hydrophobic) to polar (hydrophilic) compounds in original biomass burning aerosol
is 1:3 from Köhler Theory Analysis (Page 6, Line 133-137; Page 17, Line 451-456).

In short, the polar fraction dominates the bulk organic aerosol from biomass burning.
However, the real polar fraction in the tar balls is undefined, as tar balls belong to BBOA but
from specific burning condition with special physicochemical properties. The topic raised in
the Reviewer's question will be part of our following research topic to investigate the exact
contribution of both the polar/nonpolar matrix and to identify specific chromophore
molecules to tar ball aerosol absorption/mass. We have sent the samples of the fresh and aged
laboratory-generated polar/nonpolar tar ball for extensive molecular chemical analysis using
HPLC/PDA/HRMS (high performance liquid chromatography platform coupled to
photo-diode array and high resolution mass spectrometry detectors) and other common MS
techniques (e.g., ESI/APPI-MS, electrospray ionization/atmospheric pressure photo ionization
mass spectrometry), the results will be published in a different manuscript.

We have added the discussion of polar and nonpolar fractions contribution to tar ball
aerosols mass and absorption in the manuscript:

Page 6, Line 133-137: "The actual fractions of the polar and nonpolar compounds
contributing to the mass of ambient tar ball or biomass burning organic aerosol (BBOA) can
vary with biofuel sources, burning condition, atmospheric process, and also method/efficiency
to classify the polar and nonpolar materials from the sample (Sengupta., et al., 2018; Lin et al.,
2017, 2018; Chen and Bond, et al., 2010; Rajput et al., 2008),"

[revised manuscript text omitted]

**2. OC-EC content of fresh polar and nonpolar tar ball aerosols**

Non-refractory organic carbon (OC) and refractory elemental carbon (EC) in fresh tar ball aerosols were analyzed using a DRI Model 2015 multi-wavelength thermal/optical carbon analyzer (Desert Research Institute, Nevada, USA) with the IMPROVE_A protocol (Chow et al., 2011; Li et al., 2018). In details, fresh nonpolar and polar tar balls were collected onto pretreated quartz filters (Whatman, Mainstone, UK, baked over 450 °C for 6 hr to eliminate any contamination), a circular punch (0.8 cm in diameter) of each loaded filter including operational blank filter was taken and analyzed. Four OC fractions (OC1, OC2, OC3, and OC4 correspond to gradient cutting temperature at 140, 280, 480, and 580 °C, respectively, in a helium atmosphere), three EC fractions (EC1, EC2, EC3 with cutting temperature of 590, 780, and 840 °C, respectively, in a 2% oxygen/98% helium atmosphere), and one PC fraction (pyrolyzed carbon content determined when transmitted laser returned to its original intensity after the sample was exposed to oxygen) were determined for each sample, and OC=OC1+OC2+OC3+OC4+PC, EC=EC1+EC2+EC3-PC, total carbon (TC) equals the sum of OC and EC. The blank-corrected and normalized carbon fractions for fresh tar ball aerosols were given below:

| Tar ball | OC1 | OC2 | OC3 | OC4 | PC | EC1 | EC2 | EC3 | OC | EC |
|----------|-----|-----|-----|-----|-----|-----|-----|-----|-----|-----|
| Polar | 38.8% | 32.2% | 18.4% | 0.0% | 10.6% | 10.6% | 0.0% | 0.0% | 100.0% | 0.0% |
| Nonpolar | 28.7% | 25.8% | 16.0% | 7.7% | 21.1% | 21.7% | 0.0% | 0.0% | 99.3% | 0.7% |

It is clear EC content was almost below detection limit for both polar- and nonpolar-tar balls, the slight EC fraction in nonpolar tar ball is less than 0.7% of TC content and resides in EC1, which can be termed as non-refractory char-EC, empirically defined as EC1−PC. Char-EC is stripped from some OC under oxygen-free heating during OC/EC measurement, which has much weak absorption, and thus can be distinguished as brown carbon rather than black carbon (Andreae and Gelencsér, 2006; Arora et al., 2015; Kim et al., 2011; Han et al., 2008, 2009). Many other studies on biomass burning emissions from wildfires and domestic burning have also reported negligible EC content in tar ball aerosols (Chakrabarty et al., 2010; Tivanski et al., 2007; Hand et al., 2005; China et al., 2013).

**3. Fresh tar ball composition from HR-Tof-AMS measurement**

[Figure]

Figure S2. Fresh polar and nonpolar tar balls composition from HR-Tof-AMS measurement. Color mapping: organics-green, nitrates-blue, ammonium-yellow, chloride-purple, sulfates-red.

[revised manuscript text omitted]

d'Almeida, G. A., Koepke, P., and Shettle, E. P.: Atmospheric aerosols: global climatology and radiative characteristics, A. Deepak Publishing, Hampton, Va, 1991.

Epstein, S. A., Blair, S. L., and Nizkorodov, S. A.: Direct photolysis of α-pinene ozonolysis secondary organic aerosol: effect on particle mass and peroxide content, Environ. Sci. Technol., 48, 11251-11258, doi:10.1021/es502350u, 2014.

Hand, J. L., Malm, W., Laskin, A., Day, D., Lee, T.-b., Wang, C., Carrico, C., Carrillo, J., Cowin, J. P., and Collett, J.: Optical, physical, and chemical properties of tar balls observed during the Yosemite Aerosol Characterization Study, J. Geophys. Res.: Atmos., 110, D21210, doi:10.1029/2004JD005728, 2005.

Han, Y., Han, Z., Cao, J., Chow, J., Watson, J., An, Z., Liu, S., and Zhang, R.: Distribution and origin of carbonaceous aerosol over a rural high-mountain lake area, Northern China and its transport significance, Atmos. Environ., 42, 2405-2414, doi:10.1016/j.atmosenv.2007.12.020, 2008.

Han, Y., Lee, S., Cao, J., Ho, K., and An, Z.: Spatial distribution and seasonal variation of char-EC and soot-EC in the atmosphere over China, Atmos. Environ., 43, 6066-6073, doi:10.1016/j.atmosenv.2009.08.018, 2009.

[revised manuscript text omitted]